# Polymodal K$^+$ channel modulation contributes to dual analgesic and anti-inflammatory actions of traditional botanical medicines
Rían W. Manville[1,6], Ryan F. Yoshimura[1,6], Andriy V. Yeromin [2], Derk Hogenkamp[1], Jennifer van der Horst[3], Angel Zavala[2], Sonia Chinedu[1], Grey Arena[4], Emma Lasky[4], Mark Fisher[5], Christopher R. Tracy [5], Shivashankar Othy[2], Thomas A. Jepps [3], Michael D. Cahalan[2] & Geoffrey W. Abbott [1] ✉

Pain and inflammation contribute immeasurably to reduced quality of life, yet modern analgesic and anti-inflammatory therapeutics can cause dependence and side effects. Here, we screened 1444 plant extracts, prepared primarily from native species in California and the United States Virgin Islands, against two voltage-gated K$^+$ channels - T-cell expressed Kv1.3 and nociceptive-neuron expressed Kv7.2/7.3. A subset of extracts both inhibits Kv1.3 and activates Kv7.2/7.3 at hyperpolarized potentials, effects predicted to be anti-inflammatory and analgesic, respectively. Among the top dual hits are witch hazel and fireweed; polymodal modulation of multiple K$^+$ channel types by hydrolysable tannins contributes to their dual anti-inflammatory, analgesic actions. In silico docking and mutagenesis data suggest pore-proximal extracellular linker sequence divergence underlies opposite effects of hydrolysable tannins on different Kv1 isoforms. The findings provide molecular insights into the enduring, widespread medicinal use of witch hazel and fireweed and demonstrate a screening strategy for discovering dual anti-inflammatory, analgesic small molecules.

Many modern synthetic analgesics and anti-inflammatory drugs are highly effective but poorly tolerated or lead to dependence. The most recent wave of the opioid epidemic has been fueled primarily, in terms of deaths, by misuse of prescribed or illicitly manufactured synthetic opioids such as fentanyl or tramadol, which currently kill approximately four times as many people in the United States as heroin[1–3]. Clearly, alternative approaches are needed to combat another modern epidemic—that of chronic pain, which currently affects about 20% of the United States population[4]. In addition, given the prevalence and co-morbidity of pain with inflammatory conditions, and poor tolerance of many to steroids or NSAIDS, alternative anti-inflammatory approaches are also warranted.

In early history and pre-history, people lived from the land, collected food from a variety of challenging environments and were required to hunt and fight rival groups—each of which can lead to sores, wounds, insect and animal bites and stings, allergic or toxic reactions to plants, abrasions and broken bones. Early humans developed complex traditional medicine systems, became experts at medicinal use of plants, and discovered analgesic and anti-inflammatory properties of specific plants. These include the powerful narcotic and analgesic opium, prepared from the juice of the opium poppy (*Papaver somniferum*), one of the oldest cultivated plant species[5,6], and the bark of the willow (*Salix* sp.), which contains analgesic and anti-inflammatory compounds such as salicin[7,8].

We and others have identified voltage-gated potassium (Kv) channels as important targets for small molecules found in plants, including those used in traditional botanical medicines. In particular, channels formed from pore-forming α subunits in the Kv7 (KCNQ) subfamily have emerged as

[1]Bioelectricity Laboratory, Department of Physiology and Biophysics, School of Medicine, University of California, Irvine, CA, USA. [2]Department of Physiology and Biophysics, School of Medicine, University of California, Irvine, CA, USA. [3]Department of Biomedical Sciences, Vascular Biology Group, Panum Institute, University of Copenhagen, Copenhagen, Denmark. [4]Redwood Creek Vegetation Team, National Park Service, Sausalito, CA, USA. [5]Philip L. Boyd Deep Canyon Desert Research Center, University of California Natural Reserve System, Indian Wells, CA, USA. [6]These authors contributed equally: Rían W. Manville, Ryan F. Yoshimura. ✉e-mail: abbottg@hs.uci.edu

targets for plant metabolites that bind to sites close to the Kv7 channel pore and/or voltage sensor, to open the channel at more negative cell membrane potentials than normal. This reduces cellular excitability and is the underlying mechanism for some traditional antiepileptics, analgesics and vasorelaxant plant medicines[9–14].

While specific Kv7 isoforms are expressed in, e.g., nociceptive neurons[15], vascular smooth muscle[16–19] and in multiple regions of the brain[20–23], explaining the effects described above, Kv1.3 channels are found primarily in the immune system and are functionally very important in T-cells, as reviewed in ref. 24. Kv1.3 channels support the resting membrane potential of −50 to −60 mV in human T cells. Kv1.3 expression is highly upregulated in $T_{effector}$ cells that are chronically activated under autoimmune and inflammatory conditions. The voltage dependence of Kv1.3 provides the initial electrical driving force and sustains $Ca^{2+}$ entry that is vital for cytokine production and cell proliferation. Kv1.3 inhibition can therefore indirectly suppress $Ca^{2+}$-dependent T cell functions. Indeed, a potent and specific Kv1.3 blocker ameliorated symptoms in animal models of multiple sclerosis[25,26], and in treatment of plaque psoriasis in a limited Phase Ib human clinical trial[27]. Further work is needed to validate Kv1.3 as a therapeutic target for autoimmune diseases such as type 1 diabetes, rheumatoid arthritis and multiple sclerosis.

Plants remain a rich source for the discovery of new therapeutic small molecules to combat prevalent conditions such as pain and inflammation.

Here, we developed a library of 1444 botanical extracts, primarily from plants collected in California and the United States Virgin Islands (USVI) and screened them for their ability to open heteromeric Kv7.2/Kv7.3 channels and/or inhibit Kv1.3 channels. We validated the top dual hits in vitro, identified the underlying molecular mechanisms, and validated their effects ex vivo and in vivo.

## Results

### A subset of plant extracts open Kv7.2/7.3 and inhibit Kv1.3

High-throughput screening of 1444 plant extracts (1:50 dilution) for Kv7.2/7.3 opening and Kv1.3 inhibition revealed a large proportion of extracts exhibit one or both activities to some degree (Fig. 1A). Here, we focused only on the extracts that exhibited the highest efficacy in both actions, i.e., inhibited Kv1.3 ≥ 95% (at 2% dilution) and increased thallium flux in the Kv7.2/7.3 FLIPR assay by ≥50%; this produced a subset of 15 extracts (Fig. 1A, B). Of the 15, all but 4 have been used for indications including analgesia and/or anti-inflammation. Thus, *Chamaenerion angustifolium* (sometimes termed *Epilobium angustifolium*), which comprised 2 of the extracts in the 15, is used worldwide for a wide range of therapeutic applications, including as an anti-inflammatory and analgesic[28]. *Rosa californica* (California wild rose) has been used by Native Americans as an anti-inflammatory and analgesic[29]. *Eriogonum* sp. (wild buckwheats) have been used extensively by many Native American First Nations for a range of

**Fig. 1 | High-throughput screening of plant extracts for activity upon Kv1.3 and Kv7.2/7.3 reveals a subset with desirable dual effects.**
**A** Results of 1444-extract screen of plant extract activity against Kv1.3 and Kv7.2/3. Each point indicates screening result as the mean of a technical triplicate for an individual plant extract (1:50 extract dilution). **B** Closeup of dashed box region from (**A**) showing identities of plant extracts.

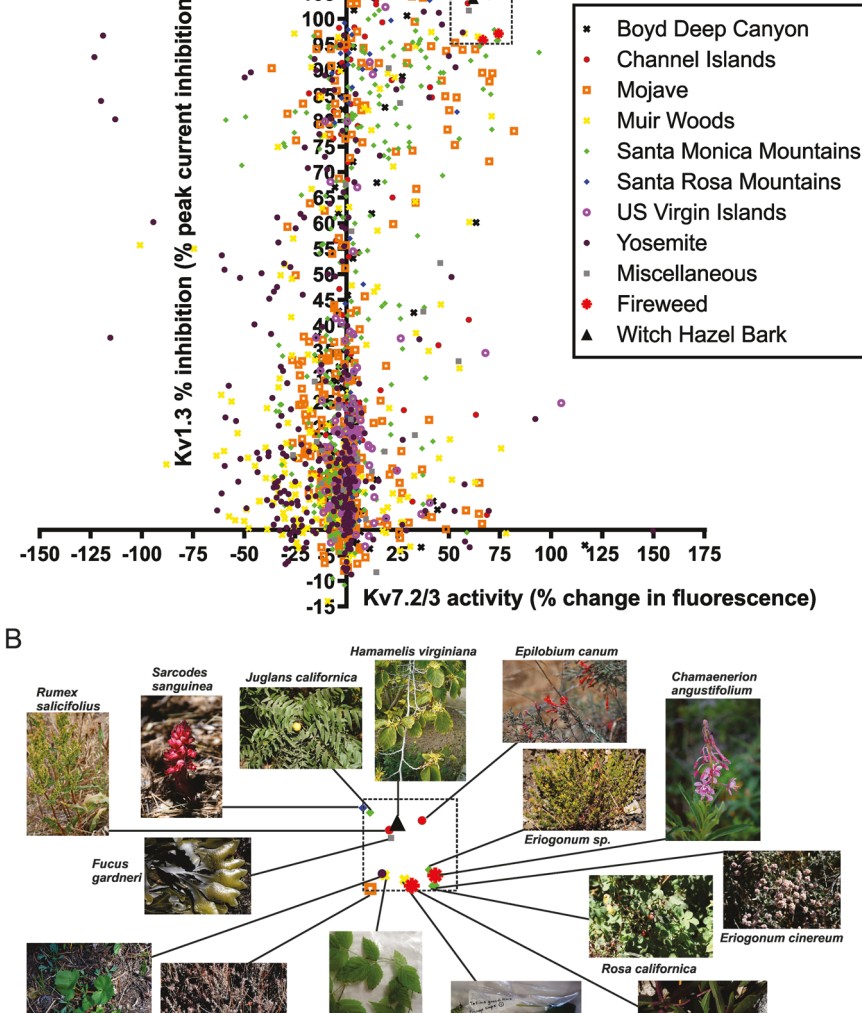

indications, including as analgesics and/or anti-inflammatory agents[30–33], as were *Hamamelis virginiana* (witch hazel)[34–36] *Saxifragaceae* sp.[29,37–39], *Fucus gardneri* (bladderwrack kelp)[40], *Rumex salicifolius* (willow dock)[41–43], *Sarcodes sanguinea* (Snow Plant)[44]. In contrast, *Tellima grandiflora* (fringecup) was used to improve appetite and as a general medicine, but we did not find analgesic or anti-inflammatory usage[29], neither did we for *Juglans californica* (California black walnut), which has been used to treat "thin blood"[45] or for *Rubus ursinus* (California Blackberry), the fruit of which is used for food and the leaves and roots of which have been used to treat gastrointestinal disorders such as diarrhea[29]. *Epilobium canum* (California fuchsia) was used by the Costanoans to treat infected cuts and sores, reduce infants' fevers, and to treat urinary tract problems, indications that suggest it may have been used for its analgesic or anti-inflammatory properties but this was not mentioned specifically[45]. Here, for further validation and mechanistic studies we focused on two that remain widely used in folk medicine—*Hamamelis virginiana* (witch hazel), which was purchased as a bark extract (Rose Mountain Herbs), and *Chamaenerion angustifolium* (fireweed, also known as rosebay willowherb or bombweed), the aerial parts of which we collected from the shores of Tenaya Lake, Yosemite National Park (Fig. 1B).

### Chemical analysis of fireweed and witch hazel bark extracts

To provide a standard for hydrolysable tannins, commercially available tannic acid was heated with acidic methanol and concentrated in vacuo before the residue was subjected to reversed-phase HPLC (MeOH/water containing 0.1% TFA), giving one major peak (retention time of 2.3 min) that co-eluted with authentic methyl gallate, as previously described[13]. Thin-layer chromatography (100% EtOAc) gave one major UV-active spot with an $R_f = 0.7$ identical to methyl gallate, as previously described[13]. Fireweed was previously reported to contain hydrolysable tannins, a group of compounds exemplified by tannic acid[28]. Here, after detecting very low levels of gallic acid or ellagic acid using mass spectrometry (MS) and tandem mass spectrometry (MS/MS), we confirmed the formation of methyl gallate after MeOH/acid treatment, using HPLC, MS and MS/MS, confirming hydrolysable tannin presence, at a concentration of 80 μg/mg extract. Using the vanillin assay[46] we detected a much lower concentration (1.6 μg/mg) of condensed tannins. We detected various additional compounds by MS and MS/MS (Supplementary Table 1).

The witch hazel extract exhibited an HPLC peak consistent with our gallic acid standard, which was confirmed by MS and daughter ion analysis. MS also indicated the presence of the mono-, di- and tri-gallate esters of a hexose (m/z 331, 483 and 635, respectively, Supplementary Table 2). Hamamelitannin, a digallate ester of a furanose (MW 484), is known to be present in witch hazel extract[47]. The mass spectrum for hamamelitannin reported in the literature[48] with negative ionization shows m/z 483 (M-H⁺) and m/z 967 (2M-H⁺) and matches the mass spectrum for witch hazel extract (Supplementary Table 2). The presence of these hydrolysable tannins conjugated with gallic acid in witch hazel extract was further confirmed based on the formation of methyl gallate when the extract (purified by HPLC to remove gallic acid) was treated with methanol and acid.

### Tannic acid effects explain contrasting fireweed effects on Kv channels

We next used manual TEVC electrophysiology to validate the ability of the fireweed extract to open Kv7.2/7.3 (Fig. 2A, B), hyperpolarize the resting membrane potential ($E_M$) of *Xenopus laevis* oocytes expressing Kv7.2/7.3 (Fig. 2C), inhibit Kv1.3 (Fig. 2F, G) and depolarize the resting membrane potential ($E_M$) of *Xenopus laevis* oocytes expressing Kv1.3 (Fig. 2H). Supporting data for all electrophysiology results are shown in Supplementary Tables 3–20.

Kv7.2/7.3 opening by fireweed involved near linearization of the tail current voltage dependence such that the channel exhibited increased constitutive activation at hyperpolarized membrane potentials but was inhibited at depolarized potentials (Fig. 2A, B). This was highly reminiscent of the effects we previously observed for tannic acid, a representative hydrolysable tannin found in certain plants, on Kv7.2/7.3 in oocytes[9].

Fireweed extract also slowed Kv7.2/3 activation by introducing a second exponential component with a tau of 2–3 s across the −20 to +40 mV range (compared to 100–500 ms across a similar voltage range for the single exponential component under control conditions) (Fig. 2D, E). As described above, fireweed contains more hydrolysable tannins, exemplified by tannic acid, than other potential candidates known to open Kv7 channels, such as gallic acid and condensed tannins. Fireweed extract inhibited Kv1.3 (Fig. 2F, G) and, as we observed for Kv7.2/3, greatly slowed activation and introduced a second, slower component of activation (Fig. 2I, J). Fireweed also almost eliminated current decay, bringing current down to the inactivated level, which suggested stabilization of the inactivated state (Fig. 2K). Effects of tannic acid on Kv1.3 were previously unreported. We found that tannic acid robustly inhibits human Kv1.3 expressed in oocytes (Fig. 2L, M), an effect associated with a shift in the midpoint voltage dependence Kv1.3 to more positive potentials (Fig. 2N). The IC$_{50}$ for Kv1.3 inhibition in oocytes is $12 \pm 1.3$ μM tannic acid (Fig. 2O), and concentrations above 10 μM also depolarized the $E_M$ of oocytes expressing Kv1.3 (Fig. 2P). The moderately slow wash-in and washout of tannic acid effects suggest a somewhat deep, but not intracellular, binding site (Fig. 2Q). Tannic acid, like fireweed, diminished currents down to the those seen during C-type inactivation in control conditions (Fig. 2L), suggesting stabilization of the inactivated state as a mode of inhibition, which we pursued further on in the study.

Thus, a single compound (tannic acid, an exemplar hydrolysable tannin) can explain the opposite effects of fireweed on Kv7.2/7.3 and Kv1.3. These effects would be predicted to make fireweed particularly effective in the late, inflammatory phase of the pain response, which we tested using the formalin paw-lick assay in mice. The fireweed extract was indeed effective at reducing pain response in vivo in the late, inflammatory phase of the formalin paw-lick assay (Fig. 3A–C). It is important to note that our data do not, however, indicate the relative contribution of each channel type to diminishing the pain response, nor do they rule out other channel types being involved. Notably, our findings correlate with prior work that supported a role for tannic acid activation of Kv7.2/3 in treating bradykinin-associated inflammatory pain[49].

### Witch hazel bark opens Kv7 channels in vitro and in vascular smooth muscle

We next studied witch hazel bark extract. At 1%, witch hazel bark extract increased Kv7.2/7.3 activity at hyperpolarized membrane potentials (Fig. 4A, B), increasing current 4.25-fold at −60 mV (Fig. 4B), inducing a −9 mV shift in $E_M$ of oocytes expressing Kv7.2/7.3 (Fig. 4C). We found previously that other medicinal bark extracts were particularly efficacious at opening the neuronal and vascular-expressed Kv7.5 channel[13]; here we found that witch hazel is also highly effective at opening Kv7.5, inducing 40% constitutive activation at −120 mV (Fig. 4D, E), >25-fold increased current at −60 mV (Fig. 4F) and a > −20 mV shift in $E_M$ of Kv7.5-expressing oocytes (Fig. 4G). This high efficacy is very consistent with our previous findings for tannic acid, which we found to (at 100 μM) induce 50% constitutive activation for Kv7.5 at −80 mV, while gallic acid was active but less efficacious[13]. One notable compound characteristic of witch hazel bark extract that we did not test previously is another hydrolysable tannin, hamamelitannin[47]. However, here we found that hamamelitannin (100 μM) was inactive on Kv7.5 expressed in oocytes (Fig. 4D–G, *lower panels*). As Kv7.5 is expressed in vascular smooth muscle and is therefore a target for vasorelaxants, we tested the effects of witch hazel bark extract on ex vivo rat mesenteric artery segments precontracted with methoxamine and found it to be an effective vasorelaxant (Fig. 4H). The relaxation elicited by witch hazel bark extract was linopirdine-sensitive, indicating that it was Kv7-dependent (Fig. 4I, J).

### Witch hazel bark inhibits Kv1.3 in oocytes and T-cells

As we found for fireweed and for tannic acid, witch hazel bark extract (1%) strongly inhibited Kv1.3 expressed heterologously in *Xenopus* oocytes (Fig. 5A, B), leading to a +8 mV depolarization of $E_M$ in oocytes expressing Kv1.3 (Fig. 5C). We also tested effects on native Kv1.3 current in activated

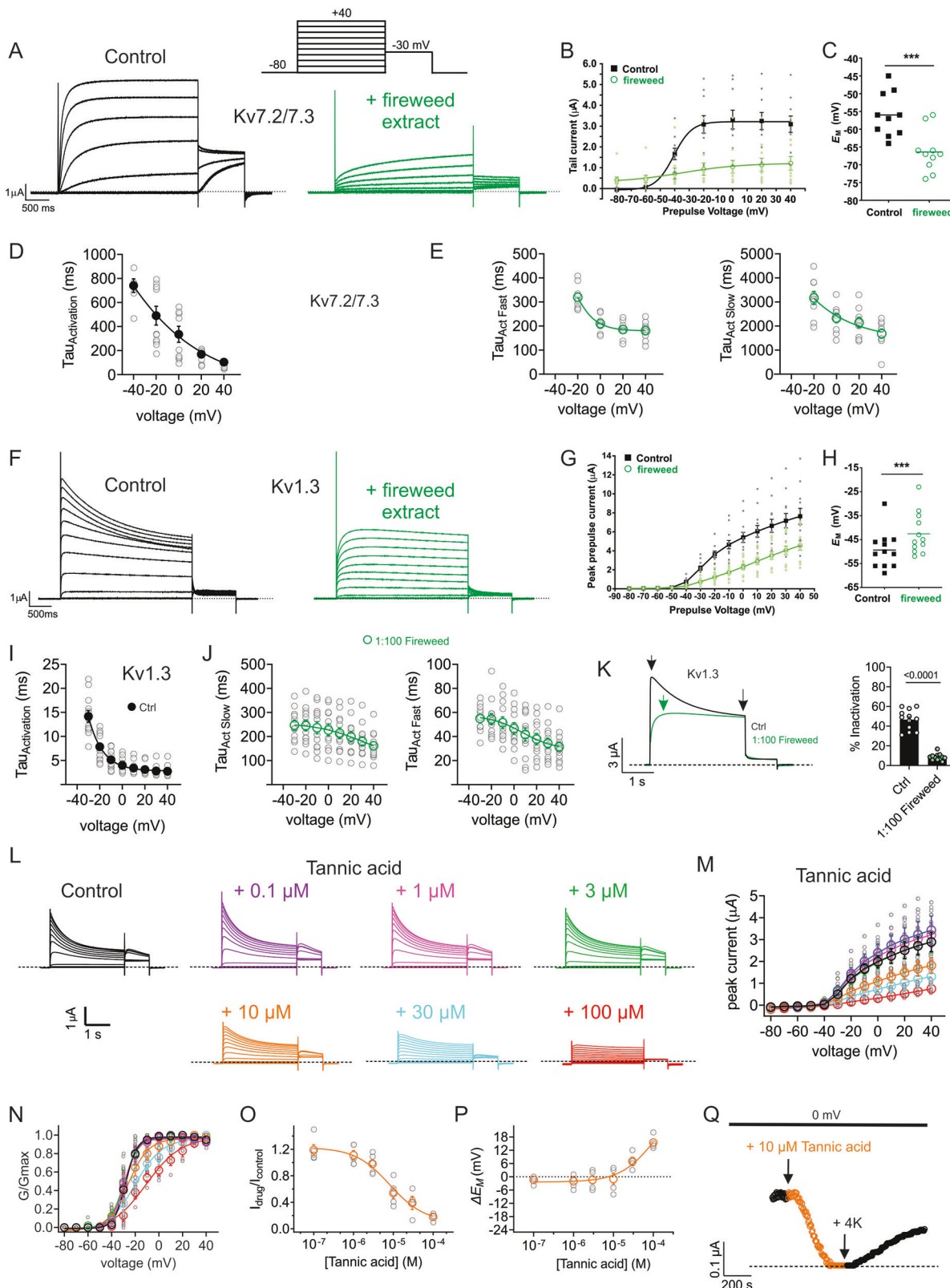

human T-cells. Witch Hazel bark extract strongly and irreversibly inhibited human T-cell Kv1.3 current (Fig. 5D, E), although construction of an accurate dose–response curve was hampered by lack of current saturation during a patch clamp protocol designed to help maintain cell health and membrane stability. The curve fit yielded an $IC_{50}$ of $6.6 \pm 0.6 \times 10^{-3}\%$ extract, equivalent to 16.5 µg/ml witch hazel extract, which is an

approximation and actually an underestimate of potency as we did not attain saturation of inhibition (Fig. 5F). Likewise, tannic acid strongly inhibited human T-cell Kv1.3 current (Fig. 5G, H) with an $IC_{50}$ of $3.3 \pm 0.2$ µM (Fig. 5I). We observed slightly more saturation of the inhibitory effect when studying tannic acid, but again this $IC_{50}$ is an approximation and an underestimate of potency because saturation was incomplete between

**Fig. 2 | Kv channel subfamily divergent effects of fireweed extract and tannic acid.**
**A** Representative traces for Kv7.2/7.3 in the absence (Control) or presence of 1%
fireweed extract. Scale bars lower left for each trace; voltage protocol upper inset;
$n = 10$. **B** Mean tail current for Kv7.2/7.3 traces as in (**A**); $n = 10$. **C** Mean unclamped
oocyte membrane potential for oocytes as in (**A**); $n = 6$–10. ***$P < 0.001$. **D** Mean
tau of activation for Kv7.2/3 Control traces as in (**A**) ($n = 9$). **E** Mean tau of activation
(fast and slow components) for Kv7.2/3 + fireweed traces as in (**A**) ($n = 10$). **F** Mean
traces for Kv1.3 in the absence (Control) or presence of 1% fireweed extract. Scale
bars lower left for each trace; $n = 12$. **G** Mean peak prepulse current for Kv1.3 traces
as in (**D**); $n = 12$. **H** Mean unclamped oocyte membrane potential for oocytes as in
(**F**); $n = 12$. ***$P < 0.001$. **I** Mean tau of activation for Kv1.3 Control traces as in (**F**)
($n = 11$–12). **J** Mean tau of activation (fast and slow components) for

Kv1.3 + fireweed traces as in (**F**) ($n = 11$–12). **K** *Left*, mean Kv1.3 traces at
+40 mV *right*, % inactivation in the absence and presence of fireweed ($n = 12$)
**L** Mean traces for Kv1.3 in the absence (Control) or presence of various con-
centrations of tannic acid as indicated; scale bars lower left; voltage protocol as in
(**A**); $n = 4$–5. **M** Mean peak prepulse current for Kv1.3 traces as in (**L**); $n = 4$–5.
**N** Mean normalized tail current (G/Gmax) for Kv1.3 traces as in (**L**); $n = 4$–5.
**O** Tannic acid dose response for Kv1.3 prepulse current inhibition from oocytes as in
(**L**), $n = 4$–5. **P** Mean unclamped oocyte $\Delta E_M$ dose response for oocytes as in (**L**);
$n = 4$–5. **Q** Representative 0 mV Kv1.3 current during tannic acid wash-in, and
washout using 4 mM K⁺ bath solution (4 K). For all panels, error bars indicate SEM.
$n$ indicates number of oocytes. Statistical comparisons by paired t-test.

**Fig. 3 | Fireweed is analgesic in the late (inflam-
matory) phase of the response to formalin injec-
tion in mice.** **A** Time spent licking the injected paw
in the early phase (acute pain response) after for-
malin injection with different dilutions of fireweed
extract versus vehicle; $n = 8$–10. **B** Time spent lick-
ing the injected paw in the late (inflammatory) phase
after formalin injection with different dilutions of
fireweed extract versus vehicle. **$P < 0.01$; $n = 8$–10.
**C** Comparison of pain response (time spent licking
injected paw) after formalin injection with different
dilutions of fireweed extract versus vehicle across the
entire experiment duration separated into 5-min
time bins. **$P < 0.01$; ***$P < 0.001$; $n = 8$–10. For all
panels, error bars indicate SEM. $n$ indicates number
of mice. Statistical comparisons by one-way or two-
way ANOVA, as appropriate.

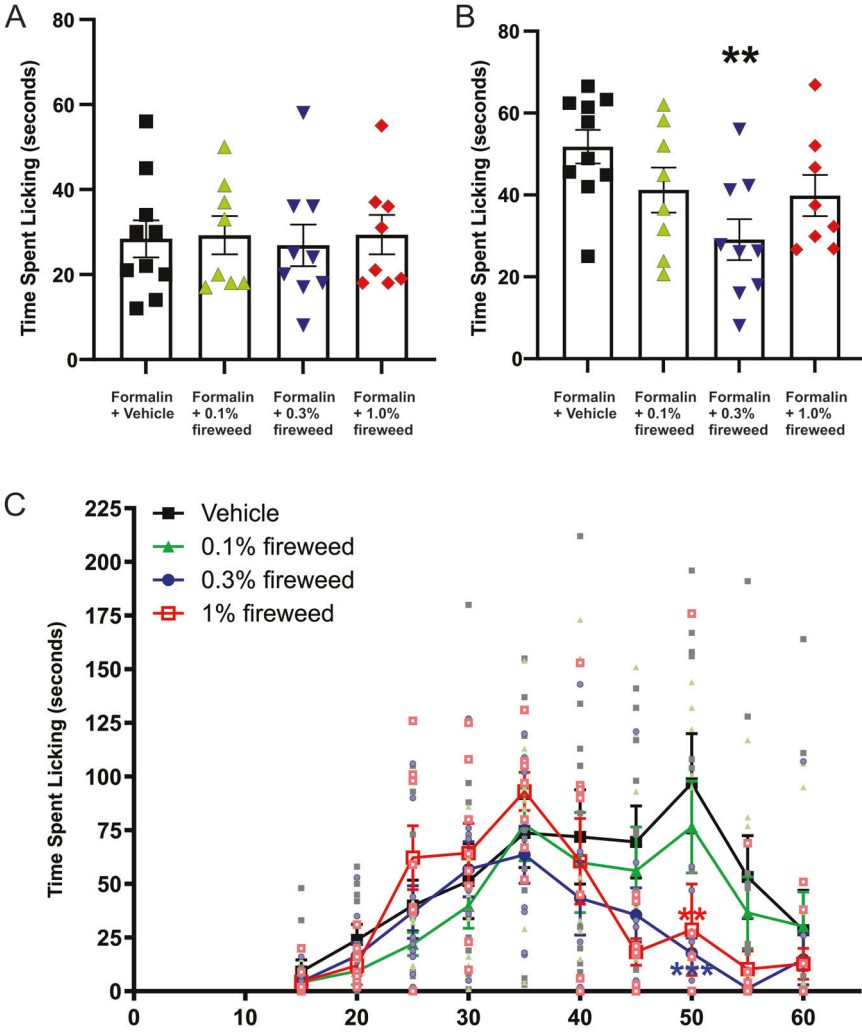

concentrations. This inhibition was partially reversible after tannic acid
washout.

**Tannic acid inhibits human CD4 T cell activation and proliferation**
To evaluate the direct effect on human T cell functions, we purified CD4
helper T cells from healthy donors and stimulated them with activating
beads (Dynabeads coated with aCD3 and aCD28 antibodies) in the presence
of various concentrations of tannic acid for 4 days (Fig. 6A). Tannic acid
inhibited T cell activation and proliferation in a concentration-dependent
manner (Fig. 6B; Supplementary Fig. 1). Both male and female T cells were
equally susceptible to the effect of Tannic acid (Supplementary Table 21).

The estimated IC$_{50}$ for limiting T cell proliferation is 35 µM (Fig. 6C, D).
Inhibition of T cell proliferation resulted in slightly diminished cell viability
(76.96 ± 8.922%) at 50 µM (Fig. 6E, F). Altogether, these data demonstrate
that tannic acid robustly inhibits human T cell activation, underscoring the
anti-inflammatory effect.

**Witch hazel bark extract and constituents activate Kv1.1 and
TREK-1**
Aside from Kv7.2/7.3 and Kv7.5, other K⁺ channels are expressed in noci-
ceptive neurons and can modulate neuronal firing and resultant pain sig-
naling, including Kv1.1[50]. Here, witch hazel bark extract reduced Kv1.1 peak

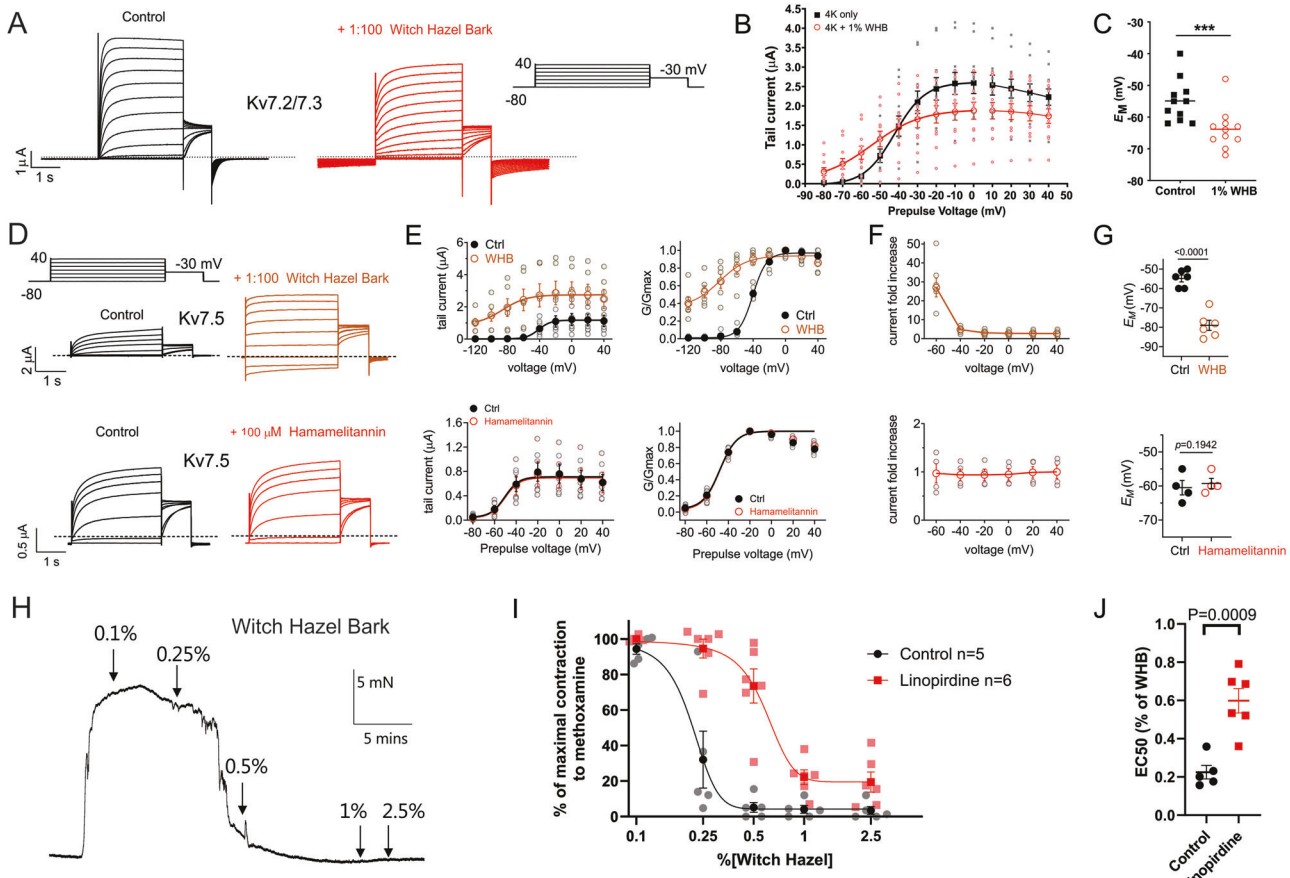

**Fig. 4 | Witch hazel bark extract activates Kv7 channels and Kv7-dependently relaxes rat mesenteric arteries. A** Exemplar traces for Kv7.2/7.3 in the absence (Control) or presence of 1% witch hazel bark extract. Scale bars lower left; voltage protocol upper right inset; $n = 11$. **B** Mean tail current for Kv7.2/7.3 traces as in (**A**); $n = 11$. **C** Mean unclamped oocyte membrane potential for oocytes as in (**A**); $n = 11$. ***$P < 0.001$. **D** Mean traces for Kv7.5 in the absence (Control) or presence of 1% witch hazel bark extract and hamamelitannin as indicated. Scale bars lower left; voltage protocol upper left inset; $n = 4–6$. **E** Mean raw (left) and normalized (right) tail current for Kv7.5 traces as in (**D**); $n = 4–6$. **F** Mean current fold increase versus voltage for traces as in (**D**); $n = 4–6$. **G** Mean unclamped oocyte membrane potential for oocytes as in (**D**); $n = 4–6$. **H** Representative myographic trace of witch hazel bark extract-mediated relaxation of pre-contracted tone (10 μM methoxamine) in a mesenteric artery from a male adult Wistar rat. **I** Mean data and **J** scatter plot with mean ± SEM of EC$_{50}$ values generated from raw data (as in **H**; $n = 5–6$) for witch hazel bark extract-mediated relaxation in the presence or absence (Control) of Kv7 inhibitor linopirdine (10 μM) in arteries from male adult Wistar rats. For all panels, error bars indicate SEM. $n$ indicates number of oocytes (**A–G**) or animals (**H–J**). Statistical comparisons by one-way ANOVA.

current at depolarized potentials (Fig. 7A) but more importantly, increased Kv1.1 activity at subthreshold membrane potentials (Fig. 7B) and was thus able to hyperpolarize (by −13 mV) the $E_M$ of Kv1.1-expressing oocytes (Fig. 7C). We recently found that tannic acid and gallic acid each negative-shift the voltage dependence of Kv1.1[50], providing a plausible molecular basis for witch hazel bark extract effects on Kv1.1, but here we also tested two other components present in witch hazel bark extract. The polyphenol catechin hydrate had negligible effects, whereas the hydrolysable tannin, hamamelitannin (100 μM) negative-shifted the voltage dependence of Kv1.1 activation (Fig. 7D–F) and hyperpolarized the $E_M$ of oocytes expressing Kv1.1 (Fig. 7G).

Primary afferent nociceptors also express members of the two-pore domain (K2P) K$^+$ channel family, and their activity helps suppress pain signaling[50]. Here, we tested a representative K2P channel, TREK-1 (KCNK2), that is expressed in nociceptive neurons, and found that it is strongly activated by 1/100 witch hazel bark extract (Fig. 7H, I), leading to a −10 mV hyperpolarization of $E_M$ in KCNK2-expressing oocytes (Fig. 7J). Tannic acid likewise robustly activated KCNK2 (Fig. 7K, L), increasing KCNK2 current >10-fold at 500 μM, with an EC$_{50}$ of 143 ± 4 μM (Fig. 7M). By activating K$^+$ channels expressed in nociceptive neurons but inhibiting the T-cell expressed Kv1.3, witch hazel bark is extremely well-suited to act as both an analgesic and an anti-inflammatory, topical botanical medicine.

## Sequence differences near the pore contribute to Kv1 isoform-specific effects of tannic acid

We previously found that tannic acid increases Kv1.1 and Kv1.2 currents by negative-shifting their voltage dependence of activation, increasing current at hyperpolarized potentials[51] but at higher concentrations also inhibits current at depolarized potentials[9]. In contrast, here we found that tannic acid strongly inhibits Kv1.3 at all potentials and positive-shifts the voltage dependence of activation. Examination of sequences from human Kv1.1, 2 and 3 revealed three regions around the pore region in which Kv1.1, Kv1.2 and Kv1.3 differ. These include an 8-residue stretch in the S5-pore helix extracellular linker (turret loop), a serine (Kv1.1, Kv1.2) to threonine (Kv1.3) switch in the C-terminal end of the pore helix, and a couplet of residues in the extracellular selectivity filter-S6 linker (Fig. 8A, B).

We used an unbiased in silico docking approach[52] to predict possible tannic acid binding poses using the pore modules for the two Kv1 isoforms for which there are high-resolution structures solved—Kv1.2 and Kv1.3. Strikingly, the second highest ranked docking pose by ΔG free energy (−11.31 kcal mol$^{-1}$) placed tannic acid close to D423 and S429 (in the turret loop) in neighboring α subunits in the Kv1.3 tetramer, and proximal to V453 in the filter-S6 couplet (Fig. 8B, C; Supplementary Fig. 2; Supplementary Table 22). The extracellular-proximal but partially protein-embedded location of tannic acid predicted from docking studies was consistent with

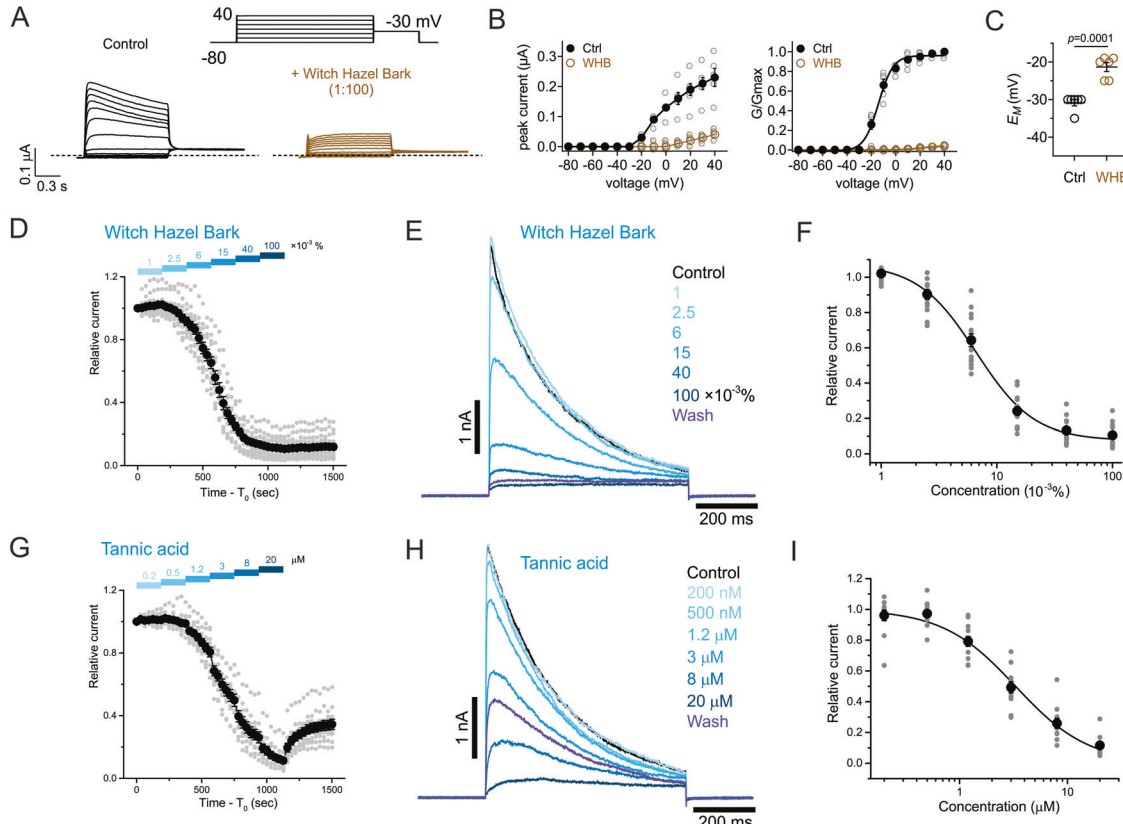

**Fig. 5 | Witch hazel bark extract and tannic acid inhibit heterologously expressed Kv1.3 and native Kv1.3 current in activated human T-cells. A** Mean traces for oocytes expressing Kv1.3 in the absence (Control) or presence of 1% witch hazel bark extract. Scale bars lower left; voltage protocol upper right inset; $n = 6$. **B** Mean peak (left) and normalized tail (right) current for Kv1.3 traces as in (**A**); $n = 6$. **C** Mean unclamped oocyte membrane potential for oocytes as in (**A**); $n = 6$. **D** Activated human T-cell Kv1.3 currents recorded in response to 600 ms test pulses to +40 mV from a holding potential of −80 mV applied every 30 s. Time course of the peak Kv1.3 current recorded upon witch hazel bark extract application ($n = 14$). For every cell the time scale was modified by subtracting $T_0$ from time, current was normalized to the current value before witch hazel bark extract application, and the data for all cells was averaged; data presented as Mean ± SEM. Bars above the time course indicate time of witch hazel bark extract application at corresponding concentrations indicated above the bars. **E** Representative traces of activated human T-cell

Kv1.3 currents at different witch hazel bark extract concentrations recorded in the same cell. After application of 0.1% extract, the extract was washed out with Ringer solution for 6 min (trace "Wash"). **F** Witch hazel bark extract dose response estimate for activated human T-cell Kv1.3 current, from traces as in (**E**), $n = 15$ cells. The curve was fitted by a modified Hill equation as indicated; $I_R = A_0 + IC_{50}^n / (IC_{50}^n + C^n)$. **G** Activated human T-cell Kv1.3 currents recorded as in (**D**), but with inhibition by tannic acid, doses as indicated above bars ($n = 11$). **H** Representative traces of Kv1.3 currents at tannic acid concentrations recorded in the same cell. After application of 20 μM tannic acid, the compound was washed out with Ringer solution for 6 min (trace "Wash"). **I** Tannic acid dose response estimate for activated human T-cell Kv1.3 current, from traces as in (**H**), $n = 12$. The curve was fitted by a modified Hill equation as indicated; $I_R = IC_{50}^n / (IC_{50}^n + C^n)$. For all panels, error bars indicate SEM. $n$ indicates number of oocytes (**A**–**C**) or T-cells (**D**–**I**). Statistical comparisons by one-way ANOVA.

its moderately slow (hundreds of ms) wash-in and washout kinetics in oocytes (Fig. 2L).

Tannic acid adopted a superficially similar binding pose in the 3rd top ranked (by DG; −10.91 kcal mol⁻¹) configuration in Kv1.2, interacting with residues of the turret loops of neighboring subunits, but in a different predicted tannic acid orientation. Further, while tannic acid was predicted to be positioned close to E353 from one subunit and P359 from the neighboring subunit (both being turret loop residues), it did not appear to be able to interact with the sidechain of Kv1.2-T383, which is in the position equivalent to V453 in the Kv1.3 filter-S6 couplet (Fig. 8B, D; Supplementary Table 23).

To test whether the predicted different binding poses of tannic acid to Kv1.2 and Kv1.3 could explain the contrasting functional effects on these two channels, we made three mutant forms of Kv1.3, introducing Kv1.2 residues into the Kv1.3 turret loop (Mutant 1), pore helix (Mutant 2) and selectivity filter-S6 linker (Mutant 3) (Fig. 9A, B). Mutant 2 was nonfunctional (Fig. 9C); the single residue in this mutant was inaccessible to tannic acid as it was deep inside the pore and was not predicted therefore to interact with tannic acid (Fig. 8). Mutants 1 and 3 were functional and exhibited less inactivation (Fig. 9B) than wild-type Kv1.3 expressed in oocytes (Fig. 2D).

Kv1.3 Mutant 1 was sensitive to tannic acid (Fig. 9B, C) but instead of positive-shifting the Kv1.3 voltage dependence of activation and depolarizing $E_M$, as we observed for wild-type Kv1.3 (Fig. 2I, K), tannic acid negative-shifted the Kv1.3 Mutant 1 midpoint voltage dependence of activation (Fig. 9D) and hyperpolarized $E_M$ (Fig. 9E), more similar to effects of tannic acid on Kv1.1 and Kv1.2[51]. Mutant 1 also reduced Kv1.3 tannic acid sensitivity threefold, consistent with it forming part of the tannic acid binding site as predicted by the docking studies (Fig. 9F).

Kv1.3 Mutant 3 was strongly inhibited by tannic acid (Fig. 9G) but exhibited neither the positive shift in voltage dependence by tannic acid that was induced in wild-type Kv1.3 (Fig. 9H; Fig. 2I) nor the depolarization of $E_M$ observed for wild-type Kv1.3 (Fig. 9I), again consistent with this residue being important for the functional effects of tannic acid, consistent with a role in the binding site. Finally, Mutant 3 did not alter the Kv1.3 tannic acid IC$_{50}$ (Fig. 9J). The mutagenesis findings are consistent with the in silico docking predictions (Fig. 8) and suggest that pore-proximal extracellular loop sequence divergences contribute to Kv1 isoform-specific differential effects of tannic acid. Importantly, as predicted, the substitutions did not prevent tannic acid binding or prevent all effects on Kv1.3, but rather made the effects of tannic acid more like those observed for Kv1.1 and Kv1.2.

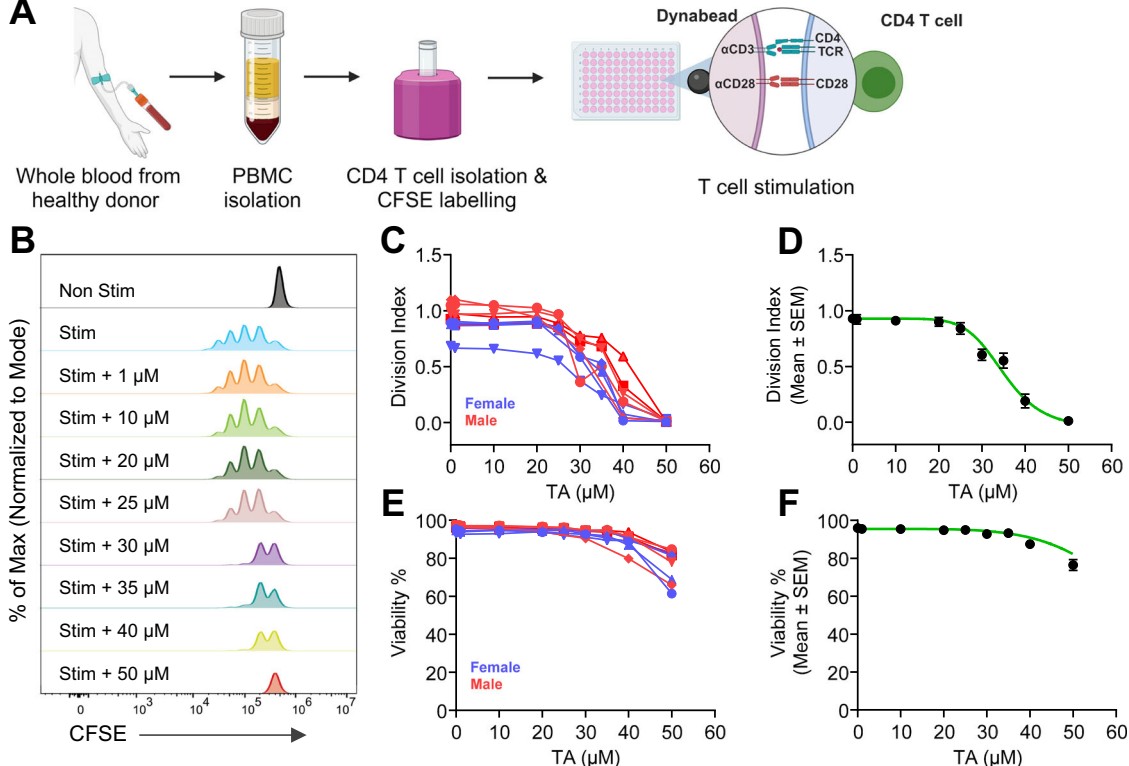

**Fig. 6 | Tannic acid limits helper T cell activation and proliferation.**
**A** Experimental design for purification of human CD4 T (helper) cells and stimulation using CD3/28 dynabeads. **B** Representative histograms (from 3–6 technical replicates of 1–2 experimental replicates from 9 individual donors) showing dilution of CFSE in CD4 T cells (gated on live) after 96 h of stimulation (Stim) in the presence of various concentrations of Tannic acid (TA). **C** Division indices of CD4$^+$ T cells at 96 h, each line represents an individual donor, and symbols represent TA

concentration (red for male donors, $n = 5$; blue for female donors, $n = 4$). **D** Symbols represent mean division indices from all donors in (**C**), (black); $n = 9$. Green line represents the dose–response curve, $IC_{50} = 35.03$ µM. **E** Viability of CD4 T cells at various TA concentrations as in (**C**). Each line represents individual donor, and symbols represent TA concentration (red for male donors, $n = 5$; blue for female donors, $n = 4$). **F** Symbols represent the mean viability of cells from all donors in (**D**), (black); $n = 9$. Green line represents the dose–response curve, $LD_{50} = 87$ µM.

## Tannic acid binds to the C-type inactivated state of Kv1.3

To test whether tannic acid inhibits activated, closed, or inactivated states, a 200-ms depolarizing pulse was applied to elicit a Kv1.3 control current and then 30 µM tannic acid was immediately added to the bath. Thereafter, consecutive 200-ms depolarizing pulses were applied every 60 s for a total of 18 pulses. Comparison of the control pre-tannic acid current (first pulse) versus the first post-tannic acid current (second pulse) revealed almost equivalent current amplitudes (Fig. 10A; *left*), suggesting tannic acid does not bind to the closed state of Kv1.3. However, upon consecutive depolarizing pulses a gradual inhibition of Kv1.3 currents developed (Fig. 10A, B, *closed circles*), indicating that tannic acid inhibition is use-dependent, and requires Kv1.3 channels to be in either the open and/ or C-type inactivated state. To confirm whether tannic acid inhibition of Kv1.3 is dependent on C-type inactivation, we employed two strategies. First, we lengthened the depolarizing pulse duration to 2 s so that more channels underwent C-type inactivation. Lengthening the pulse duration to 2 s resulted in a marked increase in Kv1.3 current inhibition, reducing the time to steady-state block to 300 ms (6 pulses) from 700 ms (13 pulses) (Fig. 10A, B; *open circles*). Second, C-type inactivation of Kv1.3 is significantly slowed by elevated [K$^+$]$_O$, which also increases the duration of the closed state of the channel[53]. Thus, we applied 30 and 100 µM tannic acid to Kv1.3 in 160 mM [K$^+$]$_O$ (Fig. 10C; *right*). and compared the inhibition to what we previously observed at 4 mM [K$^+$]$_O$ (Fig. 10C; *left*). Retardation of C-type inactivation by 160 mM [K$^+$]$_O$ abolished inhibition of Kv1.3 by 30 µM tannic acid and significantly reduced the efficacy at 100 µM (Fig. 10D). Together, these data suggest that tannic acid inhibits Kv1.3 activity primarily by binding to and stabilizing the C-type inactivated state of the channel.

## Discussion

We present the first findings from a dual-channel screen of our unique library of extracts primarily from plants collected from a range of environments in California—from desert (Mojave and Boyd Deep Canyon) to forest (Muir Woods, Yosemite, Santa Rosa Mountains), coastal canyons (Santa Monica Mountains) to alpine (Yosemite) to island chaparral and coastal sage scrub (California Channel Islands). Indigenous peoples of California, particularly before the incursion of Europeans, thrived in an incredible range of often seemingly hostile environments, expertly utilizing plants as food, medicine, and materials for fabricating homes and household objects[29,54].

We also collected tropical plants from three areas in the USVI (various locations in St. John, and Buck Island and Salt River Bay in St. Croix), some of which would have been used as food or medicine by the indigenous Taíno. Similar to indigenous peoples of North America prior to European contact, the Taíno did not have a writing system, although they used a form of proto-writing in petroglyphs[55,56]. When the remaining indigenous peoples were enslaved, killed or driven from the Virgin Islands in the sixteenth century by the Spanish, knowledge of their medicinal use of local flora was also largely lost. Hence, as for our studies in California, we collected all permitted plants and not just those for which medicinal plant usage has been documented. Africans who were brought forcibly to the Virgin Islands in slavery by Danish and English colonists developed a folk medicine system there based to some extent on similarities between the Virgin Islands flora and the flora of their African homeland, and somewhat more is known about their traditional botanical medicine practices[57,58].

Witch hazel and fireweed have been used in folk medicine practices of various cultures for centuries for their purported analgesic and

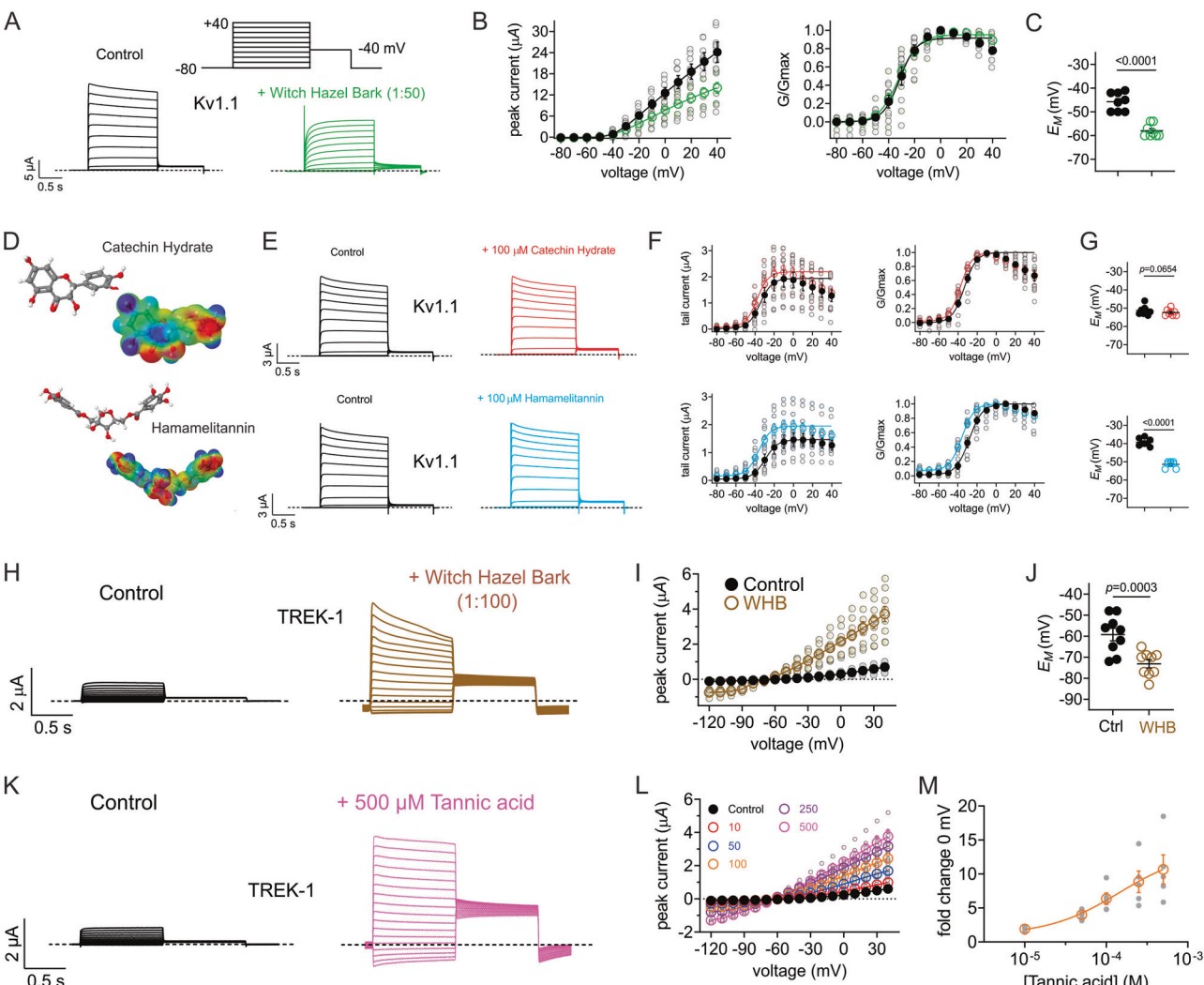

**Fig. 7 | Witch hazel bark extract and hydrolysable tannins activate Kv1.1 and TREK-1. A** Exemplar traces for Kv1.1 in the absence (Control) or presence of 1% witch hazel bark extract. Scale bars lower left; voltage protocol upper inset; $n = 8$. **B** Mean raw (left) and normalized (right) tail currents for Kv1.1 traces as in (**A**); $n = 8$. **C** Mean unclamped oocyte membrane potential for oocytes as in (**A**); $n = 8$. **D** Molecular structures (upper) and surface electrostatic plots (lower) for compounds indicated. **E** Exemplar traces for Kv1.1 in the absence (Control) or presence of witch hazel bark components indicated. Scale bars lower left; voltage protocol as in (**A**); $n = 5–6$. **F** Mean raw (left) and normalized (right) tail currents for Kv1.1 traces as in (**E**); $n = 5–6$. **G** Mean unclamped oocyte membrane potential for oocytes as in (**E**); $n = 5–6$. CH catechin hydrate, HAM hamamelitannin. **H** Exemplar traces for

TREK-1 in the absence (Control) or presence of 1% witch hazel bark extract. Scale bars lower left; voltage protocol as in (**A**) but starting at −120 mV for prepulses; $n = 9$. **I** Mean peak prepulse currents for TREK-1 traces as in (**H**); $n = 9$. WHB witch hazel bark. **J** Mean unclamped oocyte membrane potential for oocytes as in (**H**); $n = 9$. WHB witch hazel bark. **K** Exemplar traces for TREK-1 in the absence (Control) or presence of tannic acid (500 μM). Scale bars lower left; voltage protocol as in (**A**) but starting at −120 mV for prepulses; $n = 5$. **L** Mean peak prepulse currents at various tannic acid concentrations for TREK-1 traces as in (**K**); $n = 5$. **M** Mean tannic acid dose response for TREK-1 current fold increase at 0 mV for traces as in (**K**); $n = 5$. For all panels, error bars indicate SEM. $n$ indicates number of oocytes. Statistical comparisons by one-way ANOVA.

anti-inflammatory properties. Witch hazel is one of very few medicinal plants approved by the United States Food and Drug Administration as a non-prescription drug ingredient, although permissible claims are limited to relief of minor skin irritations including insect bites, minor cuts, and/or minor scrapes[59]. Native Americans used the leaves, stems and bark of the North American witch hazel, *Hamamelis virginiana*, for conditions including sore muscles, cuts, insect bites, hemorrhoids, inflammations; today, it is used to relieve hemorrhoids[60,61], diaper rash and inflammation from minor skin injuries[62]. Its use primarily as a topical agent on broken skin, which would permit greater access of tannins to subcutaneous nociceptive neurons and T cells and facilitate local concentrations of hydrolysable tannins predicted to be sufficient to exert beneficial effects on Kv7.2/3 and Kv1.3. Other groups previously found that tannic acid is effective when applied topically to improve healing of burn wounds and reduce burn pain, and also for osteoarthritis, which would require penetration through

unbroken skin[63,64]. The bark and leaves of *Hamamelis mollis* (Chinese Witch Hazel) have also been used to treat bruises, hemorrhoids, varicose veins and other skin conditions[65]. Fireweed has been used for indications as varied as gastrointestinal disorders, migraine, prostate hyperplasia, has antibacterial and antifungal effects and has been found to be efficacious in animal studies as an analgesic and anti-inflammatory therapeutic, explaining its historical use in treating sores, cuts, minor burns and ear, nose and throat inflammation[28].

The discovery that hydrolysable tannins (exemplified by tannic acid) exert polymodal actions on a variety of K⁺ channels was unexpected, and it was even more striking that tannic acid potentiates Kv1.1 and Kv1.2 activity[51] yet inhibits the closely related channel Kv1.3 (Fig. 2). Both activities are predicted to be beneficial, complementary actions that help to explain the enduring use of witch hazel and fireweed as topical analgesic/anti-inflammatory agents. It is important to note that tannic acid also has other

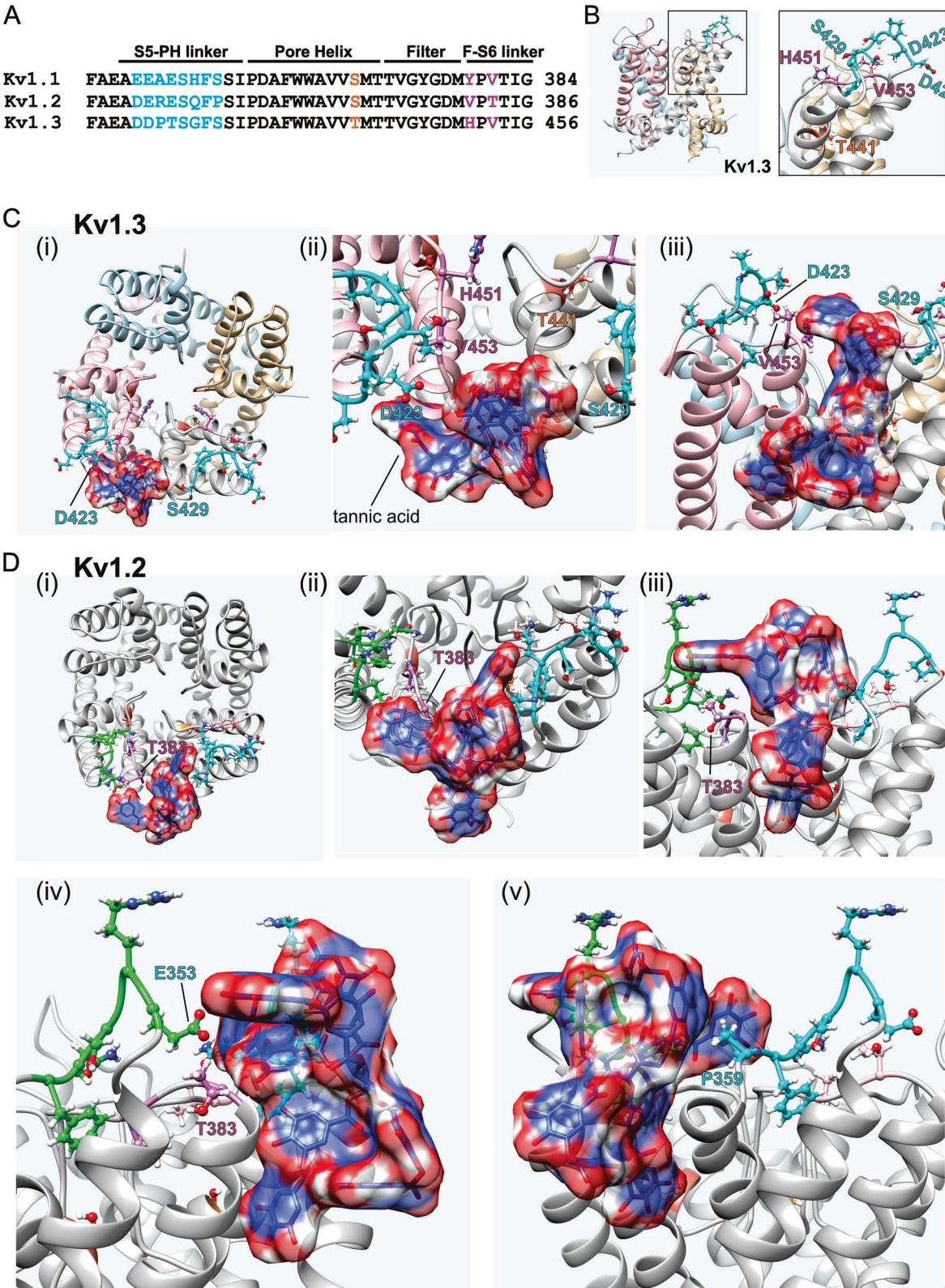

**Fig. 8 | In silico docking predicts tannic acid binding to Kv1 pore-proximal extracellular loops. A** Amino acid sequence alignment for human Kv1 isoform pore-proximal regions with divergent residues highlighted. **B** *Left*, Kv1.3 pore module structure with each of the four α subunits a different color; right, closeup of boxed region from left with divergent residues showing side chains and colored as in (**A**). **C** Kv1.3 pore module structure with tannic acid (red, white and blue) docked using SwissDock; divergent residues from two adjoining α subunits are colored as in

(**A**): (i) top view; (ii) top view closeup; (iii) side view. Enlarged view shown in Supplementary Fig. 2. **D** Kv1.2 pore module structure with tannic acid docked by SwissDock and divergent residues from two adjoining subunits colored: (i) top view; (ii) top view closeup; (iii) side view showing T383 sidechain swung away from tannic acid; (iv) side view showing tannic acid proximity to E353; (v) side view showing tannic acid proximity to P359.

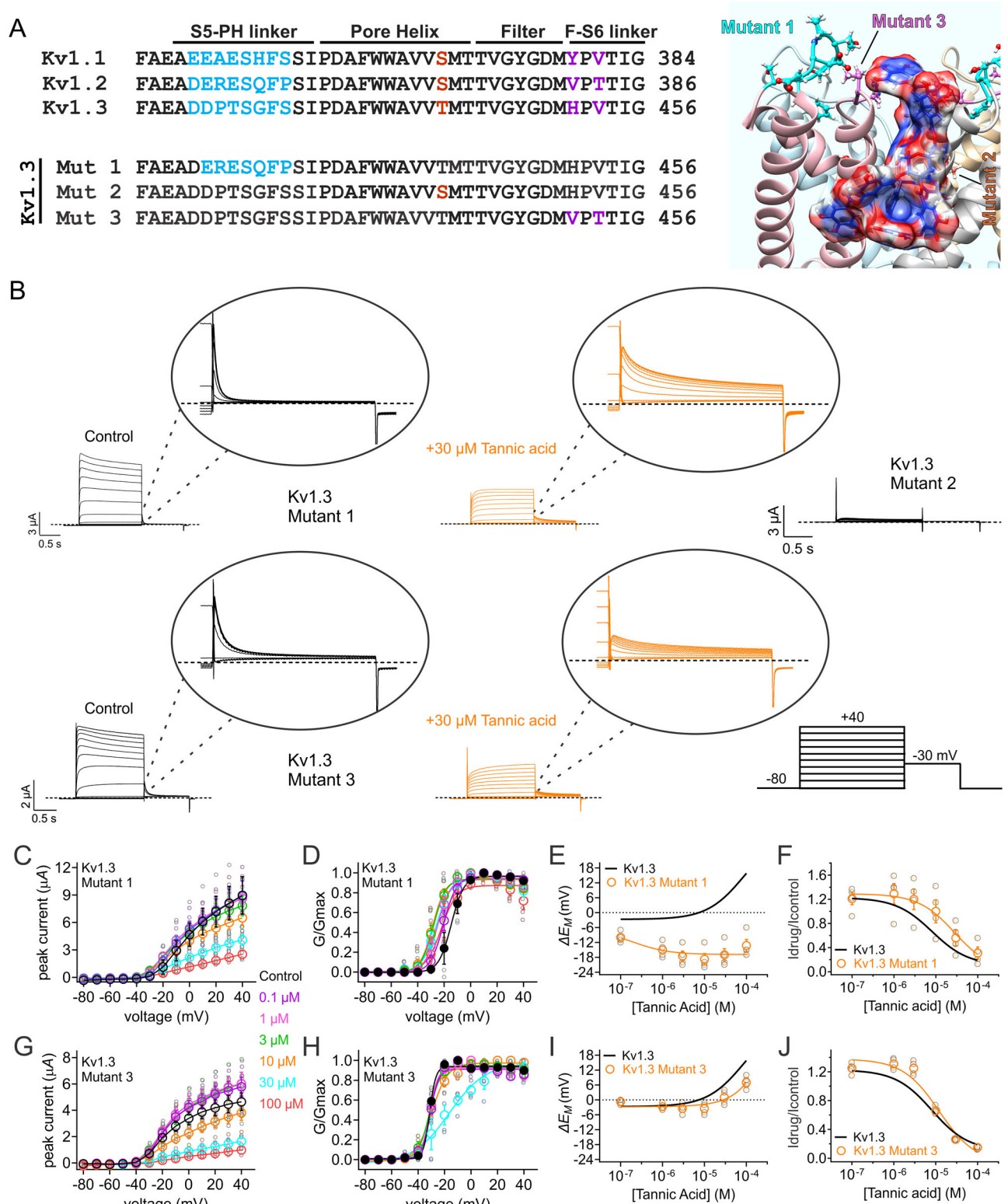

actions that almost certainly also contribute to its anti-inflammatory effects. Thus, hydrolysable tannins are known to scavenge reactive oxygen species[66], inhibit other channels, including TRPA1[67,68], and even directly inhibit cytokine synthesis via inhibition of Nuclear factor Kappa-light-chain-enhancer of activated B cells (NFKB)[69,70]. Tannic acid is a truly versatile molecule in this respect, and polymodal K$^+$ channel modulation is likely but one of its groups of effects that have anti-inflammatory and analgesic

consequences. Importantly, we demonstrate here that tannic acid inhibits the proliferation of human T cells, explaining its anti-inflammatory action, and we found this activity was similar in T cells obtained from male and female donors, suggesting that the anti-inflammatory mechanism is independent of sex. The slightly higher IC$_{50}$ in our human CD4 T cell proliferation assay compared to our electrophysiology data may be because of fetal bovine serum in the culture conditions, which is known to bind to

**Fig. 9 | Mutagenesis to investigate tannic acid binding to Kv1 pore-proximal extracellular linkers. A** *Upper left*, amino acid sequence alignment for human Kv1 isoform pore-proximal regions with divergent residues highlighted; *lower left*, Kv1.3 mutant sequences; *right*, mutant clusters colored as on left, shown on Kv1.3 structure with tannic acid docked (as in Fig. 8). **B** Mean traces for Kv1.1 mutants indicated in the absence (Control) or presence of tannic acid (30 μM); bubbles are blowups of tail currents as indicated. Scale bars lower left for each pair of traces; voltage protocol lower right inset; Mutant 2 was nonfunctional and therefore not tested with tannic acid; *n* = 4. **C** Mean peak currents for Kv1.3 Mutant 1 in tannic acid concentrations indicated (left) for traces as in (**B**); *n* = 4. **D** Mean normalized tail currents for Kv1.3 Mutant 1 currents as in (**C**); *n* = 4. **E** Mean unclamped oocyte membrane potential

for oocytes as in (**C**) compared to wild-type Kv1.3 (dashed black line, from Fig. 2P); *n* = 4. **F** Mean tannic acid inhibition dose response for Kv1.3 Mutant 1 currents as in (**C**) compared to wild-type Kv1.3 (dashed black line, from Fig. 2O); *n* = 4. **G** Mean peak currents for Kv1.3 Mutant 3 in tannic acid concentrations indicated (left) for traces as in (**B**); *n* = 4. **H** Mean normalized tail currents for Kv1.3 Mutant 3 currents as in (**G**); *n* = 4. **I** Mean unclamped oocyte membrane potential for oocytes as in (**G**) compared to wild-type Kv1.3 (dashed black line, from Fig. 2P); *n* = 4. **J** Mean tannic acid inhibition dose response for Kv1.3 Mutant 3 currents as in (**G**) compared to wild-type Kv1.3 (dashed black line, from Fig. 2O); *n* = 4. For all electrophysiology panels, error bars indicate SEM. *n* indicates number of oocytes. Statistical comparisons by one-way ANOVA.

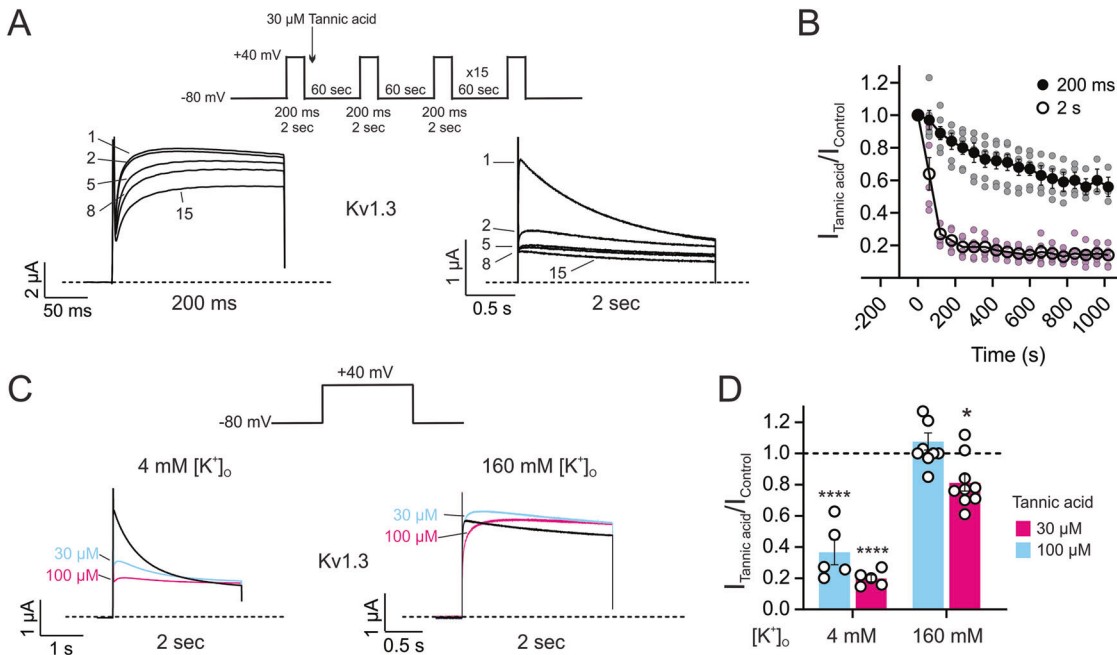

**Fig. 10 | Tannic acid binds preferentially to the C-type inactivated state of Kv1.3. A** Mean traces for the inhibition of Kv1.3 by 30 μM tannic acid during consecutive 200-ms (*left*) or 2-s (*right*) pulses. The numbers 1, 2, 5, 8 and 15 refer to pulses. Kv1.3 currents were elicited by a 200-ms or 2-s depolarizing pulse to +40 mV every 60 s for a total of 18 pulses (*inset; center*). **B** Inhibition of Kv1.3 by 30 μM tannic acid as in (**A**). Kv1.3 currents for all pulses were normalized to the control pulse pre-drug; *n* = 3–6. **C** Mean traces for the inhibition of Kv1.3 by 30 and 100 μM tannic acid in

4 mM (*left*) and 160 mM (*right*) extracellular potassium. Kv1.3 currents were elicited by a 2-s depolarizing pulse to +40 mV from a holding potential of −80 mV (*inset; center*). **D** Inhibition of Kv1.3 currents as in (**C**). Current amplitudes for Kv1.3 in 30 and 100 μM tannic acid were normalized to the control pulse pre-drug; *n* = 5–9. For all panels, error bars indicate SEM. *n* indicates number of oocytes. Statistical comparisons by one-way ANOVA.

tannic acid, but it could also indicate that other processes also contribute, as described above[71–73].

The results of our in silico docking and mutagenesis experiments support a model in which tannic acid interacts with the Kv1.3 and Kv1.2 turret loops, and that isoform-specific sequence differences in these regions and the concomitant altered tannic acid binding poses dictate the isoform-specific functional effects of tannic acid in the Kv1 family (Figs. 7, 8). Interestingly, in the recently solved structures of human Kv1.3 in complexes with immunoglobulin modulators designed to therapeutically inhibit Kv1.3 function, camelid-derived nanobodies were shown to interact not with the selectivity filter but with the Kv1.3 turret loop[74], similar (although with a different binding pose) to our data for tannic acid and in contrast with the pore-blocking binding pose of the Bovine Fab-ShK toxin chimera (MNT-002 antibody)[74]. Unlike the camelid nanobodies, which appear to accelerate and promote Kv1.3 inactivation with increasing nanobody concentrations, leading to smaller residual (non-inactivated) current[74], traces recorded with increasing concentrations of tannic acid are more consistent with retention of the residual (non-inactivated) Kv1.3 current, and together with the observed reduced tannic acid efficacy with higher extracellular K+, and increased efficacy with longer depolarizing pulses, are strongly suggestive of

tannic acid binding preferentially to the C-type inactivated state and not necessarily accelerating C-type inactivation, but rather stabilizing it to achieve inhibition (Fig. 10). Interestingly, the highly potent and selective Kv1.3 inhibitor 5-(4-phenoxybutoxy)psoralen (PAP-1), which is a derivative of 5-methoxypsoralen—isolated from *Ruta graveolens* (Common Rue) after a tea made from this plant was reported to be beneficial in multiple sclerosis—is also thought to inhibit Kv1.3 in a use-dependent manner, binding preferentially to and stabilizing the C-type inactivated state of Kv1.3 to inhibit the channel[75].

TREK-1, part of the K2P channel family that each form as dimers of α subunits each with two-pore loops, is the most intensely studied channel of its class. Similar to most K2P channels, TREK-1 contributes a "background" K+ conductance that when open tends to reduce cell excitability by driving $E_M$ toward $E_K$. It is activated by volatile general anesthetics, omega-3 polyunsaturated fatty acids, mood stabilizers and antiepileptic drugs[76]. More recently, experimental compounds have been discovered with TREK-1 activation EC50 values of 1.5 μM (BL-1249)[77,78] and 14 μM (ML335; ML402)—more potent than tannic acid (EC50 of 143 μM) but with similar or lower efficacy, depending on the expression system[79]. It is possible that there are other plant metabolites, perhaps

even other hydrolysable tannins, with potencies rivaling the synthetic compounds described above, and this will be the target for a future screen. TREK-1 is also heat-sensitive, achieving maximal activation at 37 °C, and is expressed in many neurons involved in pain perception. *Kcnk2* knockout mice are unusually sensitive to low-threshold thermal pain and exhibit exacerbated focal inflammatory response following spinal cord injury[76]. It is therefore highly plausible that TREK-1 activation by tannic acid or tannic acid-like hydrolysable tannins contributes to the analgesic and anti-inflammatory effects of witch hazel and fireweed.

We chose to focus on a small subset (15) of the 1444 plant extracts analyzed in our dual-target (Kv1.3, Kv7.2/7.3) screen for the purposes of this study, and in particular two species from the subset, but there is clearly much more to learn from the remainder of the screening data. We expect hydrolysable tannins to underlie the dual Kv1.3 inhibition/Kv7.2/7.3 activation effects of most of the 15 extracts we focused on, but perhaps not all, and it may be that some of the 15 harbor individual compounds that exhibit either but not both activities. While it is satisfying that 11 of the 15 extracts have been used historically as analgesics/anti-inflammatory therapeutics, likely primarily because of hydrolysable tannins, the utility of tannic acid-like molecules as ingested drugs is expected to be limited (although they work well as topical agents) because of some undesirable effects including gastrointestinal irritation[9,80]. However, there may be other hydrolysable tannins without this problem, and this is a topic of future research. Another future use of the screening data is to exclude plants that have both activities (Kv1.3 inhibition and Kv7.2/7.3 activation) and instead focus on plants that exhibit one or the other, with the goal of discovering active compounds that are not hydrolysable tannins, or focus on extracts that have neither effect, and screen them against K2P channels, for example. There are multiple examples of extracts that are efficacious in only one, or neither, of the actions (Fig. 1A).

In summary, using an unbiased plant extract screening approach, we found that plants used in traditional medicine practices to ease pain and inflammation beneficially target a range of K+ channels. The polymodal action of tannic acid could inform future development of more drug-like molecules that retain some or all of its impressively versatile, dual analgesic/anti-inflammatory actions, perhaps by further exploration of compounds targeting and exploiting isoform-specific sequence differences in the Kv1 channel turret loop.

## Methods
### Collection of plant samples
We collected, between 2019 and 2022, aerial parts of plants under permit from Mojave National Preserve (study # MOJA-00321), Yosemite National Park (study # YOSE-00839), Santa Monica Mountains National Recreation Area (study # SAMO-00192), Muir Woods National Monument (study # MUWO-00035), Santa Cruz Island and Santa Rosa Island (study # CHIS-0023) and Boyd Deep Canyon (Indian Wells, CA) (permit through Philip L. Boyd Deep Canyon Desert Research Center, University of California, Riverside, CA) in California, USA. In USVI, we collected from Virgin Islands National Park, St. John (study # VIIS-20001), Salt River Bay National Historical Park and Ecological Preserve, St. Croix (study # SARI-00056) and Buck Island Reef National Monument, St. Croix (study # BUIS-00103). Plant samples were collected in a manner designed to not kill the remaining plant, sealed in *Ziploc* bags (SC Johnson, Racine, WI, USA), kept cold, and frozen as soon as possible. Other plant extracts were made from plants purchased from Crimson Sage Nursery (Orleans, CA) and grown in the senior author's garden in Irvine CA, Mountain Rose Herbs (Eugene, OR) or Mother's Market (Irvine, CA). Plant samples were stored in −20 °C freezers until extraction.

### Preparation of plant extracts
We pulverized the leaves and flowers from the collected plant samples using porcelain beads agitated at high speed in a bead mill (Omni International, Kennesaw, GA, USA). Extracts were next incubated for 48 h at room temperature with periodic agitation, in 80% methanol/20% water (100 ml/5 g solid). Methanol was then evaporated off the filtered extracts before centrifugation and freezing at −20 °C.

### High-throughput screening for Kv7.2/7.3 activation
Plant extracts were applied to human Kv7.2/7.3 channels expressed in HEK293 cells (strain source, ATCC Manassas, VA; sub-strain source, ChanTest Corporation, Cleveland, OH, USA) using a FLIPR potassium assay kit and a Fluorescence Imaging Plate Reader (FLIPR$^{TETRA}$) instrument. All chemicals used in solution preparation were purchased from Sigma-Aldrich (St. Louis, MO) unless otherwise noted and were of ACS reagent grade purity or higher. Stock solutions of the positive controls were prepared in dimethyl sulfoxide (DMSO) or deionized water, aliquoted, and stored frozen. Stock solutions of the plant extracts were prepared in buffer and stored frozen after use. Test solutions of the test articles and the controls were prepared on the day of testing by diluting stock solutions into the assay buffers. The ability of each plant extract to act as an agonist of Kv7.2/7.3 channels was evaluated in a thallium (potassium ion surrogate, Molecular Devices) flux assay by Charles River Laboratories (Cleveland, OH, USA). The assay was performed with the FLIPR potassium assay kit (Molecular Devices) according to the manufacturer's instructions. For dye loading, growth media was removed and replaced with 20 µl of dye loading buffer for 60 min at room temperature. For stimulation (agonist mode): 5× (5 µL) test, vehicle, or control article solutions prepared in the stimulation buffer (K+-free buffer with 5 mM Tl+) was added to each well for ~5 min. The agonist effects of test or control articles on Kv7.2/7.3 channels were evaluated. The positive control was Flupirtine (9 concentrations). Data acquisition was performed via the ScreenWorks FLIPR control software that is supplied with the FLIPR System (MDS-AT). Data were analyzed using Microsoft Excel 2013 (Microsoft Corp., Redmond, WA). For each well, the raw kinetic data were reduced to the maximum or Area Under Curve fluorescence after subtracting bias and possibly applying the negative control correction. Reduced data were analyzed as follows:

For each assay plate, a Z′ factor and Signal Window (SW) were calculated:

Z′ factor = (([Agonist Control mean] − 3 × [Agonist Control STDEV]) − ([Vehicle Control mean] + 3 × [Vehicle Control STDEV]))/([Agonist Control mean] − [Vehicle Control mean])

SW = (([Agonist Control mean] − 3 × [Agonist Control STDEV]) − ([Vehicle Control mean] + 3 × [Vehicle Control STDEV]))/[Agonist Control STDEV]

Where the stimulation buffer was dispensed to Vehicle Control wells and a high concentration of agonist positive control was dispensed to Agonist Control wells.

Concentration–response curves were fitted to the agonist positive control.

Reduced data from test article wells were normalized to the vehicle and agonist control means on each plate and expressed as normalized percent activation:

Normalized % Activation = ([individual well RLU] − [Vehicle Control mean])/([Agonist Control mean] − [Vehicle Control mean])

Where individual well RLU are the relative light units for each well to which test article is dispensed. A significance threshold of 3 standard deviations from the vehicle control mean was calculated:

Significance Threshold = 3 × [Vehicle Control STDEV]/([Agonist Control mean] − [Vehicle Control mean])

Concentration–response curves for positive agonist controls for each plate were also conducted. The positive control results confirmed the sensitivity of the test systems to agonists. The test and control samples were prepared in the stimulation buffer (a combination of low Cl− buffer, 5 mM Tl$_2$SO$_4$ and water). The signal elicited in the presence of the positive agonist control (30 or 100 µM Flupirtine) was set to 100% activation and the signal from the vehicle (stimulation buffer) was set to 0% activation.

## High-throughput screening for Kv1.3 inhibition

Chemicals used in solution preparation were purchased from Sigma-Aldrich unless otherwise noted and were of ACS reagent grade purity or higher. Stock solutions of plant extracts and the positive controls were prepared in water and stored frozen, unless otherwise specified. Reference compound concentrations were prepared fresh daily by diluting stock solutions into a HEPES-buffered physiological saline (HB-PS) (composition in mM): NaCl, 137; KCl, 4.0; CaCl$_2$, 4.8; MgCl2, 1; HEPES, 10; Glucose, 10; pH adjusted to 7.4 with NaOH. To minimize run-down of the Kv1.3 channel currents 0.3% DMSO was added in all reference, plant extract and control solutions. The plant extracts (diluted to 2% and 0.2%) were loaded into 384-well poly-propylene compound plates and placed in the plate well of an automated patch-clamp (APC) system, SyncroPatchTM 384PE (SP384PE; Nanion Technologies, Livingston, NJ) immediately before application to Chinese Hamster Ovary (CHO) cells (strain source, ATCC Manassas, VA; sub-strain source, ChanTest Corporation, Cleveland, OH, USA) expressing human Kv1.3. Screening was conducted by Charles River Laboratories.

Extracellular buffer was loaded into the wells of the Nanion 384-well Patch Clamp (NPC-384) chips (60 µl/well). Then, cell suspension was pipetted into the wells (20 µL/well) of the NPC-384 chip. After establishment of a whole-cell patch-clamp configuration, membrane currents were recorded using the patch clamp amplifier in the SP384PE system. Plant extracts were applied to naïve cells ($n = 3$, where $n$ = the number cells/concentration). Each application consisted of addition of 40 µl of 2× concentrated test article solution to the total 80 µl of final volume of the extracellular well of the NPC-384 chip. Duration of exposure to each test article concentration was 5 min. The intracellular solution was (in mM): KCl, 70; KF, 70; MgCl2, 2; EGTA, 2.5; HEPES, 10; pH adjusted to 7.2 with KOH. In preparation for a recording session, the intracellular solution was loaded into the intracellular compartment of the NPC-384 chip. The extracellular solution was the HB-PS solution described above.

Kv1.3 channel currents were elicited using test pulses with fixed amplitudes: depolarization pulse to +20 mV amplitude, 200 ms duration from the holding potential of −90 mV. The test pulses were repeated with frequency 0.1 Hz: 3 min before (baseline) and 5 min after test articles addition. Kv1.3 channel current amplitudes were measured at the peak and at the end of the step to +20 mV. The positive control antagonist used was 4-aminopyridine, prepared as a 1 M stock in water; test concentrations were 1, 3, 10, 30, 100, 300, 1000 and 3000 µM.

## Chemical analysis of fireweed and witch hazel bark extract

**Chemicals.** (+)-Catechin hydrate, ellagic acid, gallic acid, propionyl chloride, tannic acid and vanillin were from Sigma-Aldrich; 1-O-Galloyl-beta-d-glucose was from Combi-Blocks, Inc (San Diego, CA). Quercetin was obtained from Synaptent LLC (Chicago, IL). Conc. hydrochloric acid (35–38% in water) was from Fisher. Methanol (MeOH), ethyl acetate (EtOAc) and acetonitrile (ACN) were HPLC or LC/MS grade from Fisher or VWR International. Water was 18.2 mΩ-cm from a Barnstead NANOpure Diamond™ system. Trifluoroacetic acid was obtained from EMD Millipore. To synthesize methyl gallate, a solution of gallic acid in MeOH containing sulfuric acid was refluxed overnight. Once at rt, the reaction was added to ice-water and extracted with EtOAc (3 × 25 mL). The pooled organic layers were washed twice with water, once with brine and concentrated in vacuo affording the methyl ester as a light-yellow solid. MS/MS with negative ionization mode gave m/z 183 (M-H$^+$ with daughter ions 168 and 124)[81].

**Chromatography.** Thin-layer chromatography (TLC) employed Analtech GHLF UV254 Uniplate™ silica gel plates from Miles Scientific (Newark, DE). We conducted preparative HPLC separations on a Shimadzu system consisting of two LC-8A pumps, a fraction collector (FRC-10A), a SIL-10AP auto sampler, a diode array detector (CPD-M20A) and a CBM-20A communication module. The separations employed a Waters PREP Nova-Pak® HR C18 6 µM 60 Å 40 × 100 mm reversed-phase column with a 40 × 10 mm Guard-Pak insert and a Waters PrepLC

Universal Base. We used MeOH/water gradients both containing 0.1% TFA or ACN/water gradients also with added 0.1% TFA, with fractions being collected based on their response at 254 nm.

**Mass spectroscopy.** Samples were analyzed by heated-electrospray ionization (H-ESI) in negative or positive ionization mode depending on the structure of the analyte, using a Thermo Scientific TSQ Quantum Ultra triple stage quadrupole mass spectrometer. Automatic methods for the optimization of instrument parameters were used to maximize sensitivity. Samples were analyzed by direct injection in MeOH or MeOH/water (TFA conc kept at 0.01% or less) using a syringe pump. Aqueous plant extracts were diluted 100-fold with MeOH before analysis. Gallic acid was identified from parent m/z 169 (M-H$^+$) and daughter ion m/z 125 in negative ionization mode. Ellagic acid showed m/z 301 (M-H$^+$) in negative ionization mode and its presence was confirmed by daughter ion analysis to distinguish it from quercetin, also m/z 301 (M-H$^+$).

**Transesterification analysis for hydrolysable tannins.** We conducted sample methanolysis using a 10% v/v sulfuric acid solution in MeOH. Aqueous plant extracts (1 mL) were concentrated in vacuo with a Büchi Rotavapor R-205 (water bath temp 40 °C) connected to a DryFast Ultra® pump, model 2031B-01, from Welch Rietschle Thomas. The residues obtained were dissolved in the H$_2$SO$_4$/MeOH solution (1–2 mL) and heated at 85 °C in a sealed tube under N$_2$ overnight. Once at rt, the reactions were added to ice-water/EtOAc, and then the organic layer separated and washed twice with water and conc to dryness[82]. This method was replaced with the operationally simpler method of Newsome et al.[83]. except that acetyl chloride was replaced with an equivalent molar amount of propionyl chloride. Plant extracts were concentrated to dryness as above and the residues were treated with 1–2 ml of the 2.75 M methanol–HCl solution under N$_2$. The resulting solutions were heated at 85 °C for 6 h. Once at rt, the solvent was removed in vacuo and the residues were reconstituted in MeOH and injected onto the preparative HPLC system. The methanolysis method was tested with tannic acid. Heating as above gave methyl gallate that was identified by HPLC retention time (RT), TLC R$_f$ and mass spectrum.

**Vanillin assay for condensed tannins.** The concentration of condensed tannins in the bark samples was determined using the vanillin hydrochloric acid assay. Aqueous plant extracts were diluted with an equal volume of MeOH and then 100 µL of the resulting solutions (in triplicate) diluted with 0.6 mL of a 4% w/v solution of vanillin in MeOH and 0.3 mL of conc. hydrochloric acid. The samples were vortexed briefly and then allowed to stand in the dark for 20 min. The absorbance at 490 nm was measured in a 96-well plate using a KC Junior plate reader (Bio-Tek Instruments, Vermont, USA)[46]. (+)-Catechin hydrate was used to generate a standard curve as shown in ref. 13. The condensed tannin content of the extracts is expressed as (+)-catechin equivalents (µg/mL and µg/mg). The absorbance of a blank with no (+)-catechin was subtracted from the standard curve and from the results of the plant extracts.

## Channel subunit cRNA preparation and *Xenopus laevis* oocyte injection for manual two-electrode voltage-clamp (TEVC) electrophysiology

We generated human potassium channel cRNA transcripts with the mMessage mMachine kit (Thermo Fisher Scientific, Waltham, MA, USA) according to manufacturer's instructions, after vector linearization, from cDNA sub-cloned into expression vectors (pTLNx, pXOOM and pMAX). We injected *Xenopus laevis* oocytes (Xenoocyte, Dexter, MI, USA) with the channel cRNAs (0.5–10 ng) and incubated the oocytes at 16 °C for 1–4 days prior to electrophysiological recording.

## Two-electrode voltage clamp (TEVC)

We performed TEVC at 20–23 °C with an OC-725C amplifier (Warner Instruments, Hamden, CT, USA) and pClamp10 software. Chemicals for

recording and oocyte storage solutions were from Sigma-Aldrich (St. Louis, MO). Bath solution contained (in mM): 96 NaCl, 4 KCl, 1 MgCl$_2$, 1 CaCl$_2$, 10 HEPES (pH 7.6). Pipettes were 1–2 MΩ resistance when filled with 3 M KCl. We recorded currents in response to voltage pulses between −120 mV or −80 mV and +40 mV at 10 mV intervals from a holding potential of −80 mV, to yield current–voltage relationships and examine activation kinetics. We recorded and analyzed data using Clampfit (Molecular Devices, Sunnyvale, CA, USA) and Graphpad Prism software (GraphPad, San Diego, CA, USA), stating values as mean ± SEM. We calculated the voltage dependence of activation (V$_{0.5}$) by measuring tail currents at a voltage pulse of −30 mV (Kv7) or −50 mV (Kv1), then fitted the tail current/voltage relationship with a single Boltzmann function:

$$g = \frac{(A_1 - A_2)}{\{1 + exp[V_{\frac{1}{2}} - V/Vs]\}\, y + A_2} \tag{1}$$

where $g$ is the normalized tail conductance, $A_1$ is the initial value at −∞, $A_2$ is the final value at +∞, $V_{1/2}$ is the half-maximal voltage of activation and $V_s$ the slope factor. We fitted activation and deactivation kinetics with single exponential functions.

For wild-type and mutant Kv1.3 channels we calculated conductance–voltage curves from current–voltage relationships recorded in response to voltage pulses between −80 mV to +40 mV in 10 mV intervals from a holding potential of −80 mV. The conductance was calculated by subtracting the driving force ($E_K$):

$$Ek = x\, ln\left(\frac{[K^+]i}{[K^+]o}\right) \tag{2}$$

The subsequent value was then divided by the current recorded at the relevant voltage to give the conductance corrected for $E_K$. This value was then divided against the peak conductance.

## Formalin paw-lick assay

Adult, male C57BL/6 mice (Charles River, Wilmington, MA) were group housed under a 12-h light:dark cycle and allowed access to food and water ad libitum. Mice were tested in the formalin paw-lick assay between 9 and 12 weeks of age, approved by the Institutional Animal Care and Use Committee protocol (AUP-24-095) at the University of California, Irvine.

The fireweed plant extract was prepared by diluting to 0.2%, 0.6% or 2% in sterile saline and titrating each to pH 7.4. The dilutions used for in vivo testing were estimated from the concentration required to modulate electrophysiological responses in Kv7.2/7.3 channels expressed in *Xenopus laevis* oocytes. Neutral buffered formalin (Sigma-Aldrich, St. Louis, MO) was diluted in sterile saline to a concentration of 5%. Experimental solutions were prepared by combining an equal volume of the 5% formalin solution with the diluted plant extract, resulting in a final solution containing 2.5% formalin and 0.1%, 0.3% or 1% plant extract. The vehicle control solution was prepared by combining an equal volume of formalin with sterile saline, resulting in a final solution with only 2.5% formalin. These solutions were prepared fresh daily.

Mice were habituated to the procedure room in their home cages for at least 1 h prior to testing. Either 2.5% formalin alone or 2.5% formalin with 0.1%, 0.3% or 1% plant extract was injected into the dorsal surface of the left hindpaw and then the animal was immediately placed in a large, clear polymethylpentene beaker for observation. The amount of time spent licking the injected paw was recorded in 5-min bins over 60 min by individuals blinded to the treatment received. Paw licking was separated into an early, acute phase (0–5 min post-formalin) and a late, inflammatory phase (10–60 min post-formalin).

## Mesenteric artery myography

In accordance with the methods of killing animals described in annex IV of the EU Directive 2010/63EU on the protection of animals used for scientific purposes and approved by local Animal Care and Use Committees at the University of Copenhagen (institutional approval number P21-117), male Wistar rats, 12 weeks old (Janvier Labs, France), were made unconscious by a single, percussive blow to the head. Immediately after the onset of unconsciousness, cervical dislocation was performed. Rats were group housed with regular 12-h light/dark cycles, in clear plastic containers with ad libitum access to food and water and underwent at least 1 week of habituation. After euthanasia, the intestines were removed, and third-order mesenteric arteries were dissected in ice-cold physiological saline solution containing (in mM): 121 NaCl, 2.8 KCl, 1.6 CaCl$_2$, 25 NaHCO$_3$, 1.2 KH$_2$PO$_4$, 1.2 MgSO$_4$, 0.03 EDTA and 5.5 glucose. Segments, 2 mm in length, of mesenteric artery were mounted on 40 µm stainless steel wires in a myograph (Danish Myo Technology, Aarhus, Denmark) for isometric tension recordings. The chambers of the myograph contained PSS maintained at 37 °C and aerated with 95% O$_2$/5% CO$_2$. Changes in tension were recorded by PowerLab and Chart software (ADInstruments, Oxford, UK). The arteries were equilibrated for 30 min and normalized to passive force. Artery segments were precontracted with 10 µM methoxamine (Sigma; Copenhagen, Denmark) in the absence or presence of linopirdine (10 µM) (Sigma; Copenhagen, Denmark), before application of increasing concentrations of witch hazel bark extract.

## Activated human T cells

Human CD4+ T cells from frozen PBMCs (STEMCELL Technologies) were isolated with EasySep negative isolation kit (STEMCELL Technologies, cat. #17952) according to manufacturers' protocols. After isolation cells were plated on anti-CD3 (BioLegend, clone OKT3, 2.5 µg/ml), anti-CD28 (BioLegend, clone CD28.2, 2.5 µg/ml)-coated 6-well plate dish at 1.5 × 10$^6$ cells/well in 3 ml RPMI medium (Gibco) supplemented with 10% FBS (Omega Scientific), 1% L-glutamine (Gibco), 1% non-essential amino acids (Gibco), 1% sodium pyruvate (Gibco), 1% penicillin–streptomycin–amphotericin B (Gibco) and 50 µM, β-mercaptoethanol (Sigma), 30 U/ml recombinant human IL-2 (BioLegend, cat. #589102). Cells were incubated during 48–144 h in 5% CO$_2$ + 95% O$_2$ atmosphere at 37 °C, harvested and plated on poly-lysine coated coverslips; 30–60 min later cells were used in the experiment.

## Human CD4 T cell proliferation assay

Whole blood from de-identified healthy donors, from whom informed consent was obtained, was obtained from the Institute for Clinical and Translational Science (ICTS) research Blood Donor Program at the University of California, Irvine approved and reviewed by the Institutional Review Boards of UCI (IRB protocol HS #2001-2058). All ethical regulations relevant to human research participants were followed. Peripheral blood mononuclear cells (PBMC) were isolated by density gradient separation using lymphocyte separation media (Corning, Cat# 25-072-CV) and SepMate tubes (STEMCELL Technologies, Cat# 85450). Blood was first diluted (1:3) in phosphate-buffered saline (PBS) containing 2% fetal bovine serum (FPS), then overlaid on top of lymphocyte separation media (2:1). CD4+ T cells were isolated from PBMCs using magnetic separation (EasySep Human CD4+ T cell isolation kit, STEMCELL Technologies, Cat# 17952), post isolation purity is typically >95% based on CD4 and CD3 staining. Purified CD4$^+$ T cells were labeled with 1.6 µM of CellTrace CFSE in AIM V Serum-Free Medium (**Gibco** Cat# 12055-083) for 8 min[84]. CFSE labeled CD4 T cells were stimulated with anti-CD3- and anti-CD28-coated Dynabeads (Thermo Fisher Scientific, Cat# 111310) at 1:1 ratio in a round-bottom 96-well plate at 37 °C and 5% CO$_2$ in the dark for 96 h in T cell culture medium (RPMI with 10% FCS, L-glutamine, non-essential amino acids, sodium pyruvate, β-mercaptoethanol and penn–strep with amphotericin B). On day 4 cells were stained with Fixable Viability Dye eFluor 780 (FVD-780, Thermo Fisher Scientific, Cat# 65-0865-14) in PBS for 20 min at 4 °C to identify dead cells. Cells were then washed with 1× PBS + 1% FBS twice and transferred to a round-bottom 96-deep well plate (Greiner, Cat# 07-000-119) for flow cytometry. Fluorescence intensity data were acquired using NovoCyte Quanteon (Agilent Technologies) flow

cytometer. FSC files were used for gating and analysis using FlowJo analysis software (FlowJo LLC, Ashland, OR). Gating strategy: density-based clustering was used to identify lymphocytes using forward scatter (FSC) and side scatter (SSC) bi-variate plots. Doublets were excluded using area and height of intensity FSC channel (FSC-A versus FSC-H); and single cells negative for FVD-780 were identified as live cells. CFSE+ cells within live cell gating were used to assess the dye dilution and model cell proliferation to calculate the Division Index (# Divisions/# cells at the start of culture), a measure of T cell activation.

### Patch-clamp recordings
We performed patch clamp experiments at 20–23 °C, acquiring data with an EPC9 patch clamp amplifier (HEKA), a 5 kHz sampling rate and digitally filtering at 1–2 kHz. The pipette resistance was 2–5 MΩ when filled with internal solution; pipette capacitance was completely compensated and series resistance was 80% compensated by EPC-9 circuitry; the patch seal resistance was >10 GΩ. Membrane potentials were corrected for a liquid junction potential between the pipette and bath solutions depending on internal solution composition[85]. The pipette solution composition was as follows (in mM): KF 140, $K_4EGTA$ 10, $MgCl_2$ 1.5, HEPES 10, pH = 7.2 adjusted by KOH, osmolality 300 mOsm. The external solution contained (in mM): NaCl 129, KCl 4.5, $CaCl_2$ 2, $MgCl_2$ 1, HEPES 10, glucose 10, pH = 7.4 adjusted by NaOH, osmolality 280 mOsm. The Kv1.3 currents were recorded in response to 600 ms step pulse from holding potential −80 mV to +40 mV acquired every 30 s. Once Kv1.3 current was stabilized, the tested compounds (witch hazel extract or tannic acid) diluted in bath solution at sequentially increasing concentrations were applied via local perfusion system. The recorded currents were analyzed with Patch Master (HEKA Electronics) and Origin Pro (Origin Lab).

### In silico docking
We plotted and viewed chemical structures and electrostatic surface potential using Jmol, an open-source Java viewer for chemical structures in 3D: http://jmol.org/. For in silico ligand docking predictions of binding to Kv1 channels, we performed unguided docking to predict potential binding sites, using SwissDock with CHARMM forcefields[52,86], the Kv1.2 pore module from the X-ray crystallography-derived paddle-chimera structure[87] and the cryo-EM-derived Kv1.3 structure [74]. We prepared channel structures for docking using DockPrep in UCSF Chimera (https://www.rbvi.ucsf.edu/chimera)[88], with which we also generated docking figures.

### Statistics and reproducibility
All values are expressed as mean ± SEM. Statistical differences in the formalin paw-lick assay were determined by one-way ANOVA or two-way repeated measures ANOVA, as appropriate. All others were determined by paired, two-sided $t$-test. Results were reproducible between oocyte batches or cell transfections. $n$ values are given in each figure legend.

### Data availability
Source data for Figs. 1–10 are available at: https://doi.org/10.5061/dryad.5dv41nsfc. All other data are available from the corresponding author on reasonable request.

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

## Acknowledgements

This study was supported by the National Institutes of Health, National Institute of General Medical Sciences (GM130377) and a Susan Samueli Integrative Health Institute, Samueli Scholarship to G.W.A.; an award from the Lundbeck Foundation (R323-2018-3674) to T.A.J.; and National Institutes of Health, National Institute of Allergy and Infectious Diseases R01AI168063, and U01AI160397 (to S.O.). This work was performed (in part) at the University of California Natural Reserve System, Philip L. Boyd Deep Canyon Desert Research Center, Reserve DOI: 10.21973/N3V66D. We thank Alexandra Kookootsedes and Elliot Gunnison for logistical support, collection and identification of plants in Muir Woods, Esha Kaur, Kevin Tran, Adam Buie, Kristina Mai, Catherine Tran, Deanna Tran, Henry Wu, Samy Haidar, and William Deacon (University of California, Irvine) for performing methanolic plant extractions and Catherine Tran and Francisca Okeke for conducting and/or scoring the formalin assay. We sincerely thank the iNaturalist community for their plant identification suggestions, and Kaitlyn E. Redford (University of California, Irvine) for helping with plant collection in Yosemite National Park. S.O. thanks U See I Write, a faculty writing initiative at the University of California, Irvine, which provided protected writing time to draft sections of this manuscript.

## Author contributions

G.W.A. conceived the study and coordinated plant identification; E.L., G.A., G.W.A., R.F.Y. and R.W.M. collected plants; C.R.T., E.L., G.A., G.W.A. and M.F. identified plants, provided field study expertise and/or logistical support; D.H. conducted and analyzed chemical analyses; R.W.M. and R.F.Y. conducted TEVC analysis and analyzed data; R.F.Y. coordinated and compiled the screening results; R.F.Y. and S.C. conducted pain assays; J.V.D.H. conducted myography overseen by T.A.J.; A.Z. conducted T cell assays overseen by S.O.; A.Y. conducted and analyzed T-cell electrophysiology experiments overseen by M.D.C.; G.W.A., S.O. and T.A.J. obtained funding for the project; R.W.M., T.A.J., S.O., D.H. and G.W.A. prepared the figures; G.W.A. wrote the manuscript; all authors edited the manuscript.

## Competing interests

The authors declare no competing interests.
