## [Peer Review File · Communications Biology]

Reviewers' comments:

Reviewer #1 (Remarks to the Author):

This study by Manville et al examines the effects of several natural compounds on the activity of K⁺ channels. The premise of the study is that the anti-inflammatory effects of witch hazel, fire weed, and related natural plants may arise from polymodal effects – both inhibitory and excitatory - on different classes of K⁺ channels mediated primarily through the effects of hydrolysable tannins. The concept proposed is that hydrolysable tannins in these plant extracts activate Kv7.2/7.3 channels in nociceptors to decrease C fiber activity and simultaneously decrease inflammation by inhibiting Kv1.3 channels to decrease T cell activation. The idea that tannins elicit opposing effects on select classes of cells (through differential modulation of Kv channels) is novel and interesting. However, this idea is undercut by the fact that literature shows multiple effects of hydrolysable tannins including directly inhibition of TRPA1 channels which is likely to have a major role in decreasing inflammatory pain and inhibition of inflammatory cytokines likely through affects on Nfkb which could explain the pain inhibition. These alternate affects are not discussed or brought up in the paper which is a key weakness.

Key issues:

- The anti-inflammatory effects of these natural products are attributed to Kv1.3 inhibition. While that may be partly true for T-cell activation, hydrolysable tannins in these plants also elicit very significant effects far beyond Kv1.3 inhibition including ROS scavenging, TRPA1 inhibition, and direct inhibition of cytokine synthesis through inhibition of NFkb. Parsing out the role of Kv1.3 inhibition from these other effects is difficult and this major potential confounding aspect is not discussed or mentioned at all.
- The IC50s for potentiation of Kv7 and inhibition of Kv1.3 channels appears to be in the tens of micromolar range (12 μM for tannic acid inhibition of Kv1.3, Fig2J). It is highly doubtful if such a high concentration is achieved in vivo upon consumption, or absorption through the skin. Are these concentrations in any way relevant to the physiological effects seen when used as an anti-inflammatory or analgesics?
- How is the tannic acid effect on Kv7.2/7.3 channels in nociceptors relevant to the late phase inflammatory response? If tannic acid/fire weed extract is directly activating Kv7.2/7.3 channels to hyperpolarize nociceptors and decrease C fiber activity, it should result in decreased pain response in the early phase which is not seen in Figure 3.
- Inhibition of Kv1.3 by tannic acid is very robust. The ensuing effects on T-cell function should be examined in a simple T-cell activation assay either looking at T cell proliferation or cytokine secretion which would solidify the physiological relevance of the results of

Kv1.3 inhibition on T cell function.

- Although the positive shift in voltage dependence of Kv1.3 is an interesting feature of tannic acid, this does not seem related to relevant to Kv1.3 inhibition since this is completely unaffected. (Figure 8). While the molecular docking analysis may provide some insight into the differential effects on Kv1.2 and Kv1.3, it does not inform the basis of the binding that mediates channel inhibition. It would be helpful for the paper to carry out additional molecular studies to understand the basis of inhibition.

Reviewer #2 (Remarks to the Author):

The development of drugs for treating numerous diseases from plants has a long-standing history. In this study, the authors investigated the effects of 1444 botanical extracts on Kv1.3 and Kv7.2/7.3 channels using high-throughput screening. They also validated their effects *ex vivo* and *in vivo* and studied the detailed molecular mechanism through *in silico* docking and mutagenesis. This study will provide important insights for the development of anti-inflammatory and analgesic drugs. However, some revisions are required to address specific issues.

1. In Fig 2, it appears that the fireweed extract could affect the activation of Kv1.3 and Kv7.2/7.3, as well as the inactivation of Kv1.3. It would be beneficial if the authors could add the activation and inactivation time constants results.
2. In Fig 3, although the authors demonstrated the analgesic effect of fireweed in a formalin injection-induced pain mouse model, it remains unclear whether this effect is Kv1.3- or Kv7.2/7.3-dependent.
3. In Fig 5D and 5G, the application duration of witch hazel bark and tannic acid at different concentrations is not sufficient, and the inhibitory effect has not reached stability before changing to the next concentration.

Reviewer #3 (Remarks to the Author):

Plants used in traditional medicine all over the world contain as yet unknown compounds that have potential to be directly, or after structural modifications, used in modern medicine. Finding these compounds is a challenging task. In the manuscript by Manville et al., the biological effects of fireweed and witch hazel bark extracts, as well as their active

ingredient hydrolysable tannins is described. It is shown that these extracts have a differential effect on some members of the Kv1, Kv7 and K2P subfamilies of K⁺ channels which could be useful from a pharmacological point of view. The number of plant extracts screened during this investigation is impressive. Authors successfully found extracts that were biologically active implying that the screening strategy can be applied to find new active compounds from plant extracts, in general.

The message of the manuscript is straightforward. At certain sections, addition of the quantitative information will be helpful for better estimation of findings and their pharmacological potential.

Major points.

1. The parameters of the conductance-voltage relationship are missing throughout the manuscript (Fig. 2B, Fig. 2I, Fig. 4B, Fig. 4E, Fig. 5B, Fig 6B, F et ct.). For the reader it is difficult to judge the quantitative changes. I would recommend making a table where all the parameters of the fits ($V_{1/2}$, K and amplitude A_1 , a fractional amplitude of constitutive component) will be available for the reader with corresponding statistics.

2. Line 38. Authors write “A subset of extracts both inhibited Kv1.3 and activated Kv7.2/7.3, ... Among the top dual hits were witch hazel and fireweed;.. “

The effect of fireweed extract on Kv7.2/7.3 heteromeric channels is difficult to term as “activating” in general, since it substantially inhibits the current above threshold potentials although augments currents at hyperpolarized potentials.

3. Related to point 1; how the >100-fold increase in hyperpolarized voltages (line 189) is estimated? The current at -120 without WHB is almost zero, dividing values to near zero is not an optimal way of estimation of changes. It is also difficult to understand how at -100 mV the value is increasing to 150 (fold). It will help to give the fraction of the constitutive component for control and extract cases with statistics. Ideally, the unspecific currents, estimated from water-injected oocytes, could be subtracted in calculations.

4. Lines 178-180; authors write “The fireweed extract was indeed effective at reducing pain response in vivo in the late, inflammatory phase of the formalin paw-lick assay (Figure 3A-

C)”. It is unclear which time interval is referred to as the “late (inflammatory) phase” in figure 3B and 3C. It is interesting that 1% extract is not effective in Fig. 3B, whereas lower concentration of it is analgesic (change is highly significant). Similarly, in Figure 3C a statistically significant difference is observed only for a 50 min period but from the text one has an impression that the (complete) late inflammatory phase is affected. These issues should be commented on in the manuscript.

Minor points

1. Figure 7C, 7D; Do structures represented with blue and red colors refer to electrostatic surface potentials? If yes, what is the rationale of such representation?
2. Fig. 7C is too small to clearly see the structures and the differences authors refer to in the text.
3. Fig. 2F; Y axe labeling is missing.

We thank the reviewers for their careful review and constructive comments, which we have addressed with new experiments, figures, tables, and edits to the text, discussed point-by-point below.

Reviewers' comments:

Reviewer #1 (Remarks to the Author):

This study by Manville et al examines the effects of several natural compounds on the activity of K⁺ channels. The premise of the study is that the anti-inflammatory effects of witch hazel, fire weed, and related natural plants may arise from polymodal effects – both inhibitory and excitatory - on different classes of K⁺ channels mediated primarily through the effects of hydrolysable tannins. The concept proposed is that hydrolysable tannins in these plant extracts activate Kv7.2/7.3 channels in nociceptors to decrease C fiber activity and simultaneously decrease inflammation by inhibiting Kv1.3 channels to decrease T cell activation. The idea that tannins elicit opposing effects on select classes of cells (through differential modulation of Kv channels) is novel and interesting. However, this idea is undercut by the fact that literature shows multiple effects of hydrolysable tannins including directly inhibition of TRPA1 channels which is likely to have a major role in decreasing inflammatory pain and inhibition of inflammatory cytokines likely through affects on Nfkb which could explain the pain inhibition. These alternate affects are not discussed or brought up in the paper which is a key weakness.

Key issues:

- The anti-inflammatory effects of these natural products are attributed to Kv1.3 inhibition. While that may be partly true for T-cell activation, hydrolysable tannins in these plants also elicit very significant effects far beyond Kv1.3 inhibition including ROS scavenging, TRPA1 inhibition, and direct inhibition of cytokine synthesis through inhibition of NFkb. Parsing out the role of Kv1.3 inhibition from these other effects is difficult and this major potential confounding aspect is not discussed or mentioned at all.

Thank you for raising this important issue – we have now added discussion on this (page 14). We have also softened in the title and abstract to “contributes to” rather than “underlies”, as our whole thesis is that multiple channel types are involved and TRP channels could certainly contribute as well. We have also performed new studies, using human T cells, showing that tannic acid inhibits their proliferation, an effect highly consistent with Kv1.3 inhibition (new Figure 6)

- The IC₅₀s for potentiation of Kv7 and inhibition of Kv1.3 channels appears to be in the tens of micromolar range (12 μM for tannic acid inhibition of Kv1.3, Fig2J). It is highly doubtful if such a high concentration is achieved in vivo upon consumption, or absorption through the skin. Are these concentrations in any way relevant to the physiological effects seen when used as an anti-inflammatory or analgesics?

Witch hazel and fireweed are applied topically to broken skin, cuts, sores, rashes, in which access to local nociceptive neurons and T cells would be easier than in non-broken skin. Importantly, others previously found that tannic acid is effective as a topical agent for improvement of burn wound healing and reducing burn pain, and for osteoarthritis (Halkes et al., 2001; Smith and Jacobson, 2011). We have added a note to this effect on page 14.

- How is the tannic acid effect on Kv7.2/7.3 channels in nociceptors relevant to the late phase inflammatory response? If tannic acid/fire weed extract is directly activating Kv7.2/7.3 channels to hyperpolarize nociceptors and decrease C fiber activity, it should result in decreased pain response in the early phase which is not seen in Figure 3.

The early, acute pain phase is a high bar to try to attenuate and our data suggest it is beyond what Kv7.2/3 modulation by tannic acid can accomplish, but to put this in perspective, we previously found that even 5 mg/kg morphine only attenuates early phase paw-licking by 40% (Abbott et al., 2021 Front. Physiol. 12:777057). Also, others previously suggested that tannic acid activation of KCNQ2/3 has particular utility in treating bradykinin-associated inflammatory pain (Zhang et al., 2015 Eur. J. Pharmacology). We have added discussion of this (page 7).

- Inhibition of Kv1.3 by tannic acid is very robust. The ensuing effects on T-cell function should be examined in a simple T-cell activation assay either looking at T cell proliferation or cytokine secretion which would solidify the physiological relevance of the results of Kv1.3 inhibition on T cell function.

We thank the reviewer for this suggestion on T cell proliferation. Accordingly we now include new data to demonstrate the inhibitory effect of Tannic acid on human CD4 T cell proliferation. Please refer to new Figure 6, text on page 9, lines 233-241 in the revised manuscript.

- Although the positive shift in voltage dependence of Kv1.3 is an interesting feature of tannic acid, this does not seem related to relevant to Kv1.3 inhibition since this is completely unaffected. (Figure 8). While the molecular docking analysis may provide some insight into the differential effects on Kv1.2 and Kv1.3, it does not inform the basis of the binding that mediates channel inhibition. It would be helpful for the paper to carry out additional molecular studies to understand the basis of inhibition.

In Figure 10 we show data indicative of tannic acid binding preferentially to the C-type inactivated state of Kv1.3, and we suggest this state is then stabilized. The docking predicts this is not simple pore block, and then the inactivation studies in Figure 10 show a state dependence. We did not make it clear in the original version that the state-dependent binding is expected to stabilize the inactivated state, but have made that clarification now (pages 7 and 16).

Reviewer #2 (Remarks to the Author):

The development of drugs for treating numerous diseases from plants has a long-standing history. In this study, the authors investigated the effects of 1444 botanical extracts on Kv1.3 and Kv7.2/7.3 channels using high-throughput screening. They also validated their effects *ex vivo* and *in vivo* and studied the detailed molecular mechanism through *in silico* docking and mutagenesis. This study will provide important insights for the development of anti-inflammatory and analgesic drugs. However, some revisions are required to address specific issues.

1. In Fig. 2, it appears that the fireweed extract could affect the activation of Kv1.3 and Kv7.2/3, as well as the inactivation of Kv1.3. It would be beneficial if the authors could add the activation and inactivation time constants results.

We have added the requested analyses to Figure 2 and the Supplementary Tables.

2. In Fig 3, although the authors demonstrated the analgesic effect of fireweed in a formalin injection-induced pain mouse model, it remains unclear whether this effect is Kv1.3- or Kv7.2/7.3-dependent.

Our data suggest a polymodal effect. We have now clarified this in the manuscript (pages 7-8).

3. In Fig 5D and 5G, the application duration of witch hazel bark and tannic acid at different concentrations is not sufficient, and the inhibitory effect has not reached stability before changing to the next concentration.

These studies are technically challenging as one must strike a balance between saturation and cell health during the patch clamping to create the dose response. We have added the statement that the IC50 values are an approximation (underestimate) and have explained why (page 9).

Reviewer #3 (Remarks to the Author):

Plants used in traditional medicine all over the world contain as yet unknown compounds that have potential to be directly, or after structural modifications, used in modern medicine. Finding these compounds is a challenging task. In the manuscript by Manville et al., the biological effects of fireweed and witch hazel bark extracts, as well as their active ingredient hydrolysable tannins is described. It is shown that these extracts have a differential effect on some members of the Kv1, Kv7 and K2P subfamilies of K⁺ channels which could be useful from a pharmacological point of view. The number of plant extracts screened during this investigation is impressive. Authors successfully found extracts that were biologically active implying that the screening strategy can be applied to find new active compounds from plant extracts, in general.

The message of the manuscript is straightforward. At certain sections, addition of the quantitative information will be helpful for better estimation of findings and their pharmacological potential.

Major points.

1. The parameters of the conductance-voltage relationship are missing throughout the manuscript (Fig. 2B, Fig. 2I, Fig. 4B, Fig. 4E, Fig. 5B, Fig 6B, F et ct.). For the reader it is difficult to judge the quantitative changes. I would recommend making a table where all the parameters of the fits ($V_{1/2}$, K and amplitude A_1 , a fractional amplitude of constitutive component) will be available for the reader with corresponding statistics.

We have added 18 new Supplementary Tables providing the information requested.

2. Line 38. Authors write “A subset of extracts both inhibited Kv1.3 and activated Kv7.2/7.3, ... Among the top dual hits were witch hazel and fireweed;.. “

The effect of fireweed extract on Kv7.2/7.3 heteromeric channels is difficult to term as “activating” in general, since it substantially inhibits the current above threshold potentials although augments currents at hyperpolarized potentials.

We have added the clause “at hyperpolarized potentials”. These are the most physiologically relevant potentials for regulating firing.

3. Related to point 1; how the >100-fold increase in hyperpolarized voltages (line 189) is estimated? The current at -120 without WHB is almost zero, dividing values to near zero is not an optimal way of estimation of changes. It is also difficult to understand how at -100 mV the value is increasing to 150 (fold). It will help to give the fraction of the constitutive component for control and extract cases with statistics. Ideally, the unspecific currents, estimated from water-injected oocytes, could be subtracted in calculations.

We have now changed to using current at -60 mV as the most negative voltage for the fold-change calculations, which is more accurate.

4. Lines 178-180; authors write “The fireweed extract was indeed effective at reducing pain response in vivo in the late, inflammatory phase of the formalin paw-lick assay (Figure 3A-C)”. It is unclear which time interval is referred to as the “late (inflammatory) phase” in figure 3B and 3C. It is interesting that 1% extract is not effective in Fig. 3B, whereas lower concentration of it is analgesic (change is highly significant). Similarly, in Figure 3C a statistically significant difference is observed only for a 50 min period but from the text one has an impression that the

(complete) late inflammatory phase is affected. These issues should be commented on in the manuscript.

We have added the following explanatory sentence to the Methods section: “Paw licking was separated into an early, acute phase (0–5 min post-formalin) and a late, inflammatory phase (10–60 min post-formalin).”

To address the time course question: In the time course, the only single bin that was different was the 50-minute bin, but the more relevant overall measure is the average time spent licking (which is a substitute for doing an area under the curve calculation) which clearly shows an overall reduction of licking in the inflammatory phase (Figure 3B, at 0.3%). A plausible reason for why 1% is not statistically significantly different overall is that there may be something else in the extract that may be an irritant at that concentration, which might be why there are higher levels of licking in the 25-35 minute bins. However, the licking for the 1% group from 40-60 minutes is all down by the 0.3% levels, suggesting that there may indeed be some antinociceptive effect in the "latter half of the late phase", if one were to define such a thing. We consider the time course to be interesting because if one examines the shapes of the curves, they mostly follow the same pattern (with the above noted 1% exception from 25-35 min) for the first 35 minutes and then they diverge with the vehicle and 0.1% staying at higher levels of licking and the 0.3% and 1% moving to lower levels of licking.

Minor points

1. Figure 7C, 7D; Do structures represented with blue and red colors refer to electrostatic surface potentials? If yes, what is the rationale of such representation?

The coloring of the tannic acid is solely to make it stand out clearly and does not denote surface potential.

2. Fig. 7C is too small to clearly see the structures and the differences authors refer to in the text.

Fig 7 already fills the page, so we have added a new Supplementary Figure 1 with an enlarged Figure 7C.

3. Fig. 2F; Y axis labeling is missing.

We have added the axis label.

REVIEWERS' COMMENTS:

Reviewer #1 (Remarks to the Author):

This revised manuscript addresses the main concerns I had about the effects of witch hazel, fire weed, and related natural plants on the activity of K⁺ channel. Although I believe that the in silico docking analysis is not terribly meaningful for understanding the mechanism of the drugs, overall, the authors have addressed my concerns. I have no further comments.

Reviewer #3 (Remarks to the Author):

The authors have addressed all of my comments and concerns thoroughly. I don't have any more questions.

Reviewer #4 (Remarks to the Author):

The authors have addressed my concerns in the revised manuscript. I believe the manuscript can be accepted for publication. Thank you!